# Deconstructing Guidance: A Semantic Hierarchy for Precise Diffusion Model Editing

**Wootaek Jeong**[1][*]    **Junghyo Sohn**[1][*]    **Jee Seok Yoon**[2]    **Heung-Il Suk**[1][†]

[1]Department of Artificial Intelligence, Korea University        [2]SK hynix

wtjeong@korea.ac.kr    jhsohn0633@korea.ac.kr
jeeseok.yoon@sk.com    hisuk@korea.ac.kr

## Abstract

Text-guided image editing requires more than prompt following—it demands a principled understanding of what to modify versus what to preserve. We investigate the internal guidance mechanism of diffusion models and reveal that the guidance signal follows a structured semantic hierarchy. We formalize this insight as the Semantic Scale Hypothesis: the magnitude of the guidance difference vector ($\Delta \epsilon$) directly encodes the semantic scale of edits. Crucially, this phenomenon is theoretically grounded in Tweedie's formula, which links score prediction to the variance of the underlying data distribution. Low-variance regions, such as objects, yield large-magnitude differences corresponding to structural edits, whereas high-variance regions, such as backgrounds, yield small-magnitude differences corresponding to stylistic adjustments. Building on this principle, we introduce Prism-Edit, a training-free, plug-and-play module that decomposes the guidance signal into semantic layers, enabling selective and interpretable control. Extensive experiments—spanning direct visualization of the semantic hierarchy, generalization across foundation models, and integration with state-of-the-art editors—demonstrate that Prism-Edit achieves precise, robust, and controllable editing. Our findings establish semantic scale as a foundational axis for understanding and advancing diffusion-based image editing.

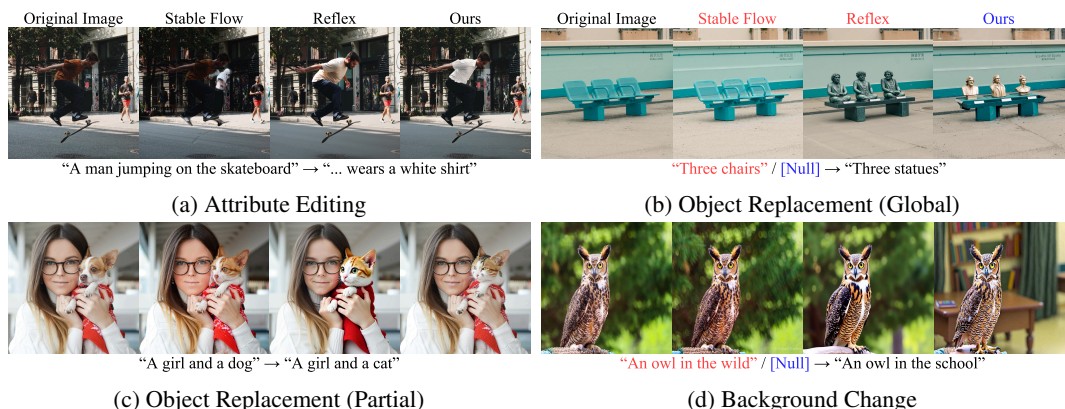

"A man jumping on the skateboard" → "... wears a white shirt"

(a) Attribute Editing

"Three chairs" / [Null] → "Three statues"

(b) Object Replacement (Global)

"A girl and a dog" → "A girl and a cat"

(c) Object Replacement (Partial)

"An owl in the wild" / [Null] → "An owl in the school"

(d) Background Change

Figure 1: Prism-Edit achieves competitive precision across diverse editing tasks. By decomposing the guidance signal, our method prevents common failure modes like semantic leakage (a) and content degradation (b, c), while excelling at the challenging task of background modification (d).

---

[*]Equal contribution
[†]Corresponding author

## 1 INTRODUCTION

The ability to sculpt our visual world through natural language is a central ambition of artificial intelligence. Recent advancements in text-to-image diffusion models (Rombach et al., 2022; Esser et al., 2024; Labs, 2024) have brought this vision closer to reality, largely powered by Classifier-Free Guidance (CFG) (Ho & Salimans, 2021). Yet despite their remarkable success, current editing methods suffer from a persistent weakness: **background regions are notoriously difficult to modify**, while object-centric edits succeed more reliably (Figure 1). For instance, a command to move an owl from "the wild" to "a school" often fails to convincingly alter the scene or inadvertently degrades the subject.

Prior approaches have mainly attacked this problem through heuristic spatial controls, asking where an edit should occur. These techniques often involve manipulating cross-attention maps (Hertz et al., 2023; Cao et al., 2023; Kim et al., 2025) or using the guidance difference vector to generate a spatial mask that separates the image into edit and preserve zones (Couairon et al., 2023).

In contrast, we argue the true bottleneck lies in *how* the guidance signal itself is structured. We show that the guidance difference vector $\Delta\epsilon$, central to CFG, is not random noise but the *gradient of a log-likelihood ratio*, whose expected magnitude is governed by local Fisher information density. This framing reveals a fundamental statistical law: **objects, being information-dense, naturally yield strong guidance, whereas backgrounds, being information-sparse, yield weak guidance**. We call this principle the **Semantic Scale Hypothesis**, which reinterprets background editing failure as an information-theoretic inevitability rather than an incidental flaw of prior methods.

Building on this insight, we propose **Prism-Edit**, a training-free, model-agnostic technique that decomposes the guidance signal into semantic layers and selectively amplifies the weak, low-information components corresponding to backgrounds. As previewed in Figure 1, this enables precise object edits while, for the first time, delivering robust and controllable background modifications. Our contributions are threefold:

1. **Semantic Scale Hypothesis:** We formalize a new principle that connects guidance magnitude to Fisher information, providing the first theoretical explanation for the persistent difficulty of background editing.

2. **Prism-Edit:** A simple, training-free, and model-agnostic method that operationalizes this principle by amplifying low-information signals.

3. **Extensive Validation:** We validate our hypothesis and method across multiple foundation models, showing consistent gains over state-of-the-art editors, especially for challenging background edits.

## 2 RELATED WORK

Text-guided image editing with diffusion models (Ho et al., 2020; Song et al., 2021; Ramesh et al., 2022; Esser et al., 2024; Labs, 2024), initiated by methods like SDEdit (Meng et al., 2022), has predominantly focused on spatial control—determining **"WHERE"** to apply edits. This paradigm includes techniques like manipulating attention maps (Tumanyan et al., 2023; Hertz et al., 2023; Cao et al., 2023) or refining sampling trajectories (Brack et al., 2024) to localize changes. Notably, DiffEdit (Couairon et al., 2023) pioneered using the guidance difference vector, $\Delta\epsilon$, to automatically generate a spatial mask. However, this still interprets the signal spatially, partitioning the image into a binary "edit" versus "preserve" zone. We provide a detailed comparison highlighting the fundamental difference between DiffEdit's masking strategy and our gradient modulation approach in Appendix C.7.

Our work poses a complementary question: **"HOW"** should an edit be applied? We shift the focus from the signal's location to its intrinsic semantic nature. We posit that the magnitude of $\Delta\epsilon$ is not merely a spatial indicator, but a rich signal encoding a semantic hierarchy. Instead of creating a binary mask, our method decomposes this signal into distinct semantic layers (e.g., object structure, style/background). This enables a more expressive, disentangled form of control by modulating the guidance signal's intrinsic semantic structure rather than just its spatial application.

And our work builds upon the standard framework of text-to-image diffusion models (Ho et al., 2020) and Classifier-Free Guidance (CFG) (Ho & Salimans, 2021). During sampling, CFG steers the generation process by extrapolating from an unconditional noise prediction $\epsilon_\theta(\mathbf{x}_t, \emptyset)$ towards a conditional prediction $\epsilon_\theta(\mathbf{x}_t, c)$. Our analysis focuses on the core of this mechanism: the **Guidance Difference Vector**, $\Delta\epsilon = \epsilon_\theta(\mathbf{x}_t, c_{\text{target}}) - \epsilon_\theta(\mathbf{x}_t, c_{\text{source}})$, which represents the model's perceived direction to transform a source concept into a target. We hypothesize that the magnitude of this vector, $\|\Delta\epsilon\|$, is a structured semantic signal.

## 3 THEORETICAL FOUNDATION: GUIDANCE AS A GRADIENT FIELD

Our central claim, the Semantic Scale Hypothesis, is not merely an empirical observation but appears to be a direct consequence of the statistical principles governing diffusion models. This section provides a first-principles derivation, showing that the guidance difference vector $\Delta\epsilon$ acts as a gradient field of a log-likelihood ratio, whose magnitude is intrinsically linked to the local information density of the image. The temporal evolution in Figure 2 provides strong empirical support for this derived theory.

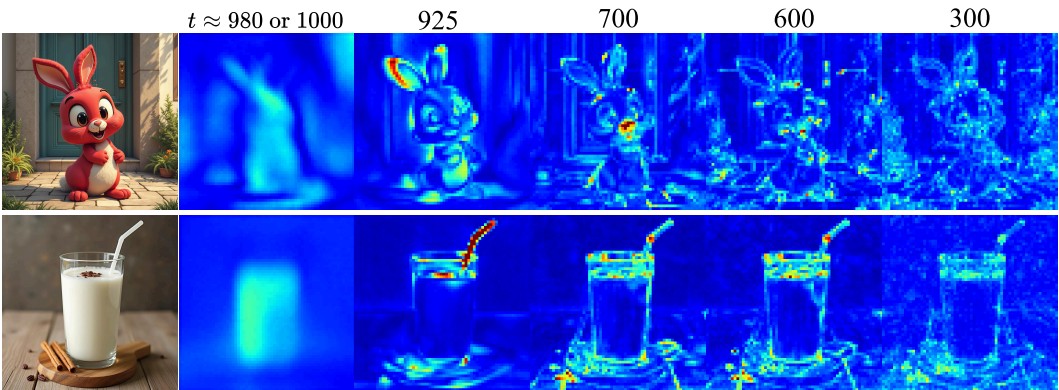

Figure 2: Temporal evolution of the guidance difference vector, $\|\Delta\epsilon\|$, during a standard generation trajectory. High-magnitude signals, encoding object structure, dominate in the early-to-mid timesteps before diminishing, while background regions remain consistently low-magnitude throughout. This provides strong empirical support for the Semantic Scale Hypothesis.

### 3.1 THE GUIDANCE DIFFERENCE AS A LOG-LIKELIHOOD RATIO GRADIENT

The foundation of a diffusion model is its score function, $\nabla_{\mathbf{x}_t} \log p(\mathbf{x}_t \mid c)$, which points in the direction of maximal increase in data likelihood. With the standard $\epsilon$-parameterization, the predicted noise $\epsilon_\theta(\mathbf{x}_t, c)$ is proportional to this score. Classifier-Free Guidance steers the generation by taking the difference between a conditional and an unconditional prediction. The core of any edit, however, is the difference between two conditional predictions (source $c_1$ and target $c_2$). This guidance difference vector, $\Delta\epsilon$, is therefore proportional to the difference between two scores:

$$\Delta\epsilon(\mathbf{x}_t; c_1, c_2) \propto \nabla_{\mathbf{x}_t} \log p(\mathbf{x}_t|c_2) - \nabla_{\mathbf{x}_t} \log p(\mathbf{x}_t|c_1). \tag{1}$$

By the properties of logarithms, this simplifies to the gradient of a single scalar field: the log-likelihood ratio between the target and source conditions.

$$\Delta\epsilon(\mathbf{x}_t; c_1, c_2) \propto \nabla_{\mathbf{x}_t} \log \frac{p(\mathbf{x}_t|c_2)}{p(\mathbf{x}_t|c_1)}. \tag{2}$$

This reframes the guidance vector: it is not just a directional hint, but a vector field that points "uphill" on the surface of how much more likely the noisy image $\mathbf{x}_t$ is under the target condition versus the source condition. The magnitude $\|\Delta\epsilon\|$ thus reflects the steepness of this likelihood ratio landscape.

## 3.2 INFORMATION DENSITY AND POSTERIOR CERTAINTY

The steepness of the landscape in Eq. 2 is determined by the model's "certainty" about the underlying clean image $\mathbf{x}_0$. This certainty is directly related to the local information density of the image content, a principle linked to the model's posterior via Tweedie's formula (Efron, 2011).

- **Structured Regions (e.g., objects):** These areas are characterized by high information density (edges, textures, recognizable forms). Given a noisy patch from an object, the model has strong priors, leading to a sharp posterior distribution $p(\mathbf{x}_0 \mid \mathbf{x}_t, c)$ with low variance. The model is "certain" about what should be there.
- **Smooth Regions (e.g., backgrounds):** These areas have low information density (smooth gradients, skies, walls). The model's posterior is flat, with high variance, as many clean signals could have resulted in the same noisy patch. The model is "uncertain."

A sharp, low-variance posterior means that a small change in the condition (from $c_1$ to $c_2$) can cause a dramatic shift in the posterior mean $\mathbb{E}[\mathbf{x}_0 \mid \mathbf{x}_t, c]$. Conversely, for a flat, high-variance posterior, the same conditional change results in a much smaller shift.

## 3.3 SEMANTIC SCALE AS A CONSEQUENCE OF INFORMATION DENSITY

We can now connect these principles. The magnitude $\|\Delta\boldsymbol{\epsilon}\|$ is proportional to the posterior mean shift $\|\Delta\mu_t\|$:

$$\|\Delta\boldsymbol{\epsilon}(\mathbf{x}_t; c_1, c_2)\| \;\propto\; \frac{\|\Delta\mu_t\|}{\sigma_t}, \quad \Delta\mu_t := \mathbb{E}[\mathbf{x}_0 \mid \mathbf{x}_t, c_2] - \mathbb{E}[\mathbf{x}_0 \mid \mathbf{x}_t, c_1]. \tag{3}$$

This proportionality follows from the relationship between the $\epsilon$-parameterization and the posterior mean derived from Tweedie's formula (a brief proof sketch is provided in Appendix A for clarity). As established in Section 4.2, edits concerning high-information, low-variance regions (objects) induce large posterior shifts ($\|\Delta\mu_t\|$), resulting in a large-magnitude $\|\Delta\boldsymbol{\epsilon}\|$. Edits concerning low-information, high-variance regions (backgrounds, styles) induce small shifts, resulting in a small-magnitude $\|\Delta\boldsymbol{\epsilon}\|$. Therefore, we interpret the Semantic Scale Hypothesis as a natural consequence of applying information-theoretic principles to the score-matching objective. Large-magnitude guidance is not just correlated with objects; it appears to be the mathematical result of the model being more "certain" and "opinionated" about these information-dense regions. Prism-Edit is the first method to leverage this insight, reframing editing not as a masking problem, but as a principled signal processing challenge: separating and amplifying semantically crucial components based on their information-theoretic signature.

## 3.4 CLOSED-FORM BOUNDS UNDER GAUSSIAN POSTERIOR APPROXIMATION

The proportionality in Eq. 3 can be made quantitatively precise by adopting a local Gaussian approximation of the posterior:

$$p(\mathbf{x}_0 \mid \mathbf{x}_t, c_i) \approx \mathcal{N}(\boldsymbol{\mu}_{c_i}, \boldsymbol{\Sigma}_{c_i}) \qquad (i \in \{1, 2\}).$$

Under this approximation, the mean shift is $\Delta\boldsymbol{\mu}_t := \boldsymbol{\mu}_{c_2} - \boldsymbol{\mu}_{c_1}$ and Eq. 3 suggests $\|\Delta\boldsymbol{\epsilon}\| \propto \|\Delta\boldsymbol{\mu}_t\|/\sigma_t$. We now upper/lower bound $\|\Delta\boldsymbol{\epsilon}\|^2$ in terms of closed-form divergences between Gaussians, which separates *mean-shift* and *covariance-mismatch* effects.

**Theorem 1** (KL-based bound for guidance magnitude). *Let $d$ be the dimensionality and define the Gaussian KL divergence*

$$D_{\mathrm{KL}}(\mathcal{N}(\boldsymbol{\mu}_{c_1}, \boldsymbol{\Sigma}_{c_1}) \,\|\, \mathcal{N}(\boldsymbol{\mu}_{c_2}, \boldsymbol{\Sigma}_{c_2})) = \tfrac{1}{2}\left[\mathrm{tr}(\boldsymbol{\Sigma}_{c_2}^{-1}\boldsymbol{\Sigma}_{c_1}) + (\Delta\boldsymbol{\mu}_t)^{\top}\boldsymbol{\Sigma}_{c_2}^{-1}\Delta\boldsymbol{\mu}_t - d + \log\frac{\det\boldsymbol{\Sigma}_{c_2}}{\det\boldsymbol{\Sigma}_{c_1}}\right].$$

*Let $\lambda_{\max}(\boldsymbol{\Sigma})$ and $\lambda_{\min}(\boldsymbol{\Sigma})$ denote the largest and smallest eigenvalues of a symmetric positive definite matrix $\boldsymbol{\Sigma}$, respectively. Then, for any $t$,*

$$\|\Delta\boldsymbol{\epsilon}\|^2 \leq \frac{\lambda_{\max}(\boldsymbol{\Sigma}_{c_2})}{\sigma_t^2}\Big\{2\,D_{\mathrm{KL}}(\mathcal{N}(\boldsymbol{\mu}_{c_1}, \boldsymbol{\Sigma}_{c_1}) \,\|\, \mathcal{N}(\boldsymbol{\mu}_{c_2}, \boldsymbol{\Sigma}_{c_2}))$$

$$- \underbrace{\Big[\mathrm{tr}(\boldsymbol{\Sigma}_{c_2}^{-1}\boldsymbol{\Sigma}_{c_1}) - d - \log\det(\boldsymbol{\Sigma}_{c_2}^{-1}\boldsymbol{\Sigma}_{c_1})\Big]}_{:= \Psi(\boldsymbol{\Sigma}_{c_1}, \boldsymbol{\Sigma}_{c_2})}\Big\}. \tag{4}$$

*and symmetrically with $(c_1, c_2)$ swapped:*

$$\|\Delta\boldsymbol{\epsilon}\|^2 \; \leq \; \frac{\lambda_{\max}(\boldsymbol{\Sigma}_{c_1})}{\sigma_t^2}\Big\{2\, D_{\mathrm{KL}}(\mathcal{N}(\boldsymbol{\mu}_{c_2}, \boldsymbol{\Sigma}_{c_2}) \,\|\, \mathcal{N}(\boldsymbol{\mu}_{c_1}, \boldsymbol{\Sigma}_{c_1})) - \Psi(\boldsymbol{\Sigma}_{c_2}, \boldsymbol{\Sigma}_{c_1})\Big\}. \qquad (5)$$

*Moreover, the following lower bounds hold:*

$$\|\Delta\boldsymbol{\epsilon}\|^2 \; \geq \; \frac{\lambda_{\min}(\boldsymbol{\Sigma}_{c_2})}{\sigma_t^2}\Big\{2\, D_{\mathrm{KL}}(\mathcal{N}(\boldsymbol{\mu}_{c_1}, \boldsymbol{\Sigma}_{c_1}) \,\|\, \mathcal{N}(\boldsymbol{\mu}_{c_2}, \boldsymbol{\Sigma}_{c_2})) - \Psi(\boldsymbol{\Sigma}_{c_1}, \boldsymbol{\Sigma}_{c_2})\Big\}, \qquad (6)$$

*and analogously with $(c_1, c_2)$ swapped.*

**Interpretation.** The term $\Psi(\boldsymbol{\Sigma}_{c_1}, \boldsymbol{\Sigma}_{c_2}) := \mathrm{tr}(\boldsymbol{\Sigma}_{c_2}^{-1}\boldsymbol{\Sigma}_{c_1}) - d - \log\det(\boldsymbol{\Sigma}_{c_2}^{-1}\boldsymbol{\Sigma}_{c_1}) \geq 0$ quantifies the *covariance mismatch* (it vanishes iff $\boldsymbol{\Sigma}_{c_1} = \boldsymbol{\Sigma}_{c_2}$). Hence the bound cleanly separates (i) the *mean-shift* captured by the KL divergence and (ii) the *uncertainty gap* captured by $\Psi$.

**Corollary 1** (Equal-covariance simplification)**.** *If $\boldsymbol{\Sigma}_{c_1} = \boldsymbol{\Sigma}_{c_2} = \boldsymbol{\Sigma}$, then $\Psi = 0$ and*

$$\frac{2\,\lambda_{\min}(\boldsymbol{\Sigma})}{\sigma_t^2}\, D_{\mathrm{KL}}(\mathcal{N}(\boldsymbol{\mu}_{c_1}, \boldsymbol{\Sigma}) \,\|\, \mathcal{N}(\boldsymbol{\mu}_{c_2}, \boldsymbol{\Sigma})) \; \leq \; \|\Delta\boldsymbol{\epsilon}\|^2 \; \leq \; \frac{2\,\lambda_{\max}(\boldsymbol{\Sigma})}{\sigma_t^2}\, D_{\mathrm{KL}}(\mathcal{N}(\boldsymbol{\mu}_{c_1}, \boldsymbol{\Sigma}) \,\|\, \mathcal{N}(\boldsymbol{\mu}_{c_2}, \boldsymbol{\Sigma})) \,.$$

*Thus larger guidance magnitude is driven either by a larger mean shift (object-level changes) or by smaller posterior variance (higher certainty), making the object/background gap visible in both* mean *and* covariance *channels.*

**Connection to Fisher divergence.** From Eq. A.2, taking an expectation w.r.t. any reference density $q(\mathbf{x}_t)$ gives

$$\mathbb{E}_q\big[\|\Delta\boldsymbol{\epsilon}\|^2\big] \; = \; \sigma_t^2\, \mathbb{E}_q\Big[\big\|\nabla_{\mathbf{x}_t}\log p(\mathbf{x}_t\,|\,c_2) - \nabla_{\mathbf{x}_t}\log p(\mathbf{x}_t\,|\,c_1)\big\|^2\Big] \; = \; \sigma_t^2\, \mathcal{F}_q\big(p(\cdot\,|\,c_2),\, p(\cdot\,|\,c_1)\big),$$

the (generalized) Fisher divergence between the two conditionals under $q$. When $q = p(\cdot\,|\,c_2)$ (or $c_1$), $\mathcal{F}_q$ reduces to the standard Fisher divergence; for Gaussian pairs, $\mathcal{F}_q$ admits a closed form, revealing the same mean/covariance decomposition as in Theorem 1.

### 3.5 SEMANTIC SCALE AS INFORMATION-THEORETIC NECESSITY

In summary,

$$\|\Delta\boldsymbol{\epsilon}\|^2 \; \propto \; \text{local Fisher information density.}$$

Objects, being information-dense, inevitably yield large guidance, while backgrounds, being information-sparse, yield vanishing signals. Thus the **Semantic Scale Hypothesis** is not an empirical artifact but a direct corollary of score matching and Fisher information theory, explaining background editing failure as a statistical necessity.

## 4 METHOD: A PRINCIPLED FRAMEWORK FOR DISENTANGLED EDITING

Building on our theoretical foundation, we introduce **Prism-Edit**, a framework designed to operationalize the Semantic Scale Hypothesis for precise, disentangled image editing. Unlike methods that rely on external parsers or attention manipulation, Prism-Edit derives its control signals directly from the model's internal generation dynamics. As illustrated in Figure 3, our approach is a two-stage process: (1) principled extraction of a multi-layered **Semantic Map**, and (2) disentangled application of edits via one of two complementary modalities.

### 4.1 STAGE 1: SEMANTIC MAP EXTRACTION

Section 3 established that the guidance magnitude $\|\Delta\boldsymbol{\epsilon}\|$ scales with the posterior mean shift normalized by the posterior variance, i.e., with the local Fisher information density. This implies that absolute magnitudes are not directly comparable across timesteps or samples: regions with high variance (backgrounds) systematically appear weak, even when the underlying semantic change is substantial.

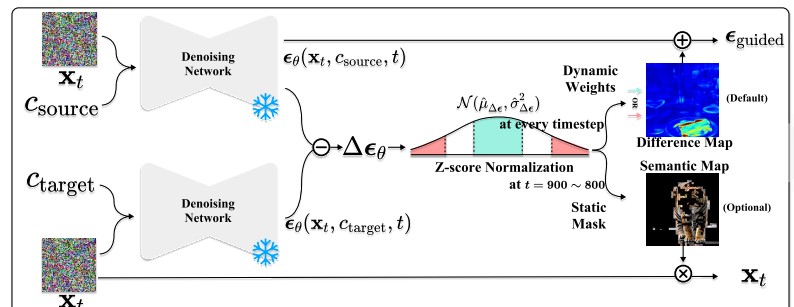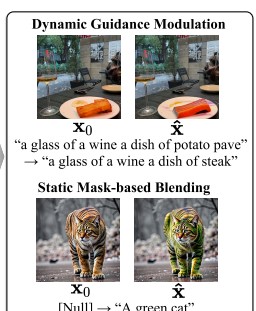

Figure 3: **Overall Prism-Edit framework.** The Semantic Map is extracted from early denoising (left), and applied during sampling via *dynamic guidance modulation* (default) or *static mask blending* (optional). Region-specific scaling enables strong background edits without destabilizing objects.

To compensate for this Fisher information imbalance, we adopt a $\sigma$-normalized thresholding scheme. Specifically, we probe a narrow, high-noise window (e.g., $t \in [900, 800]$ for a 1000-step schedule). As detailed in Appendix C.6, this specific interval was selected based on empirical analysis showing it maximizes semantic coverage while retaining structural plasticity, unlike later timesteps, which become overly rigid. We then compute an averaged guidance difference:

$$\overline{\Delta\boldsymbol{\epsilon}} = \frac{1}{N_{\text{probe}}} \sum_{i=1}^{N_{\text{probe}}} \Delta\boldsymbol{\epsilon}_{t_i}, \qquad M_{\text{sem}} = \frac{|\overline{\Delta\boldsymbol{\epsilon}}| - \mu_{|\overline{\Delta\boldsymbol{\epsilon}}|}}{\sigma_{|\overline{\Delta\boldsymbol{\epsilon}}|}}, \tag{7}$$

where $N_{\text{probe}}$ denote the number of probed timesteps in the chosen window. As predicted in Corollary 1, the raw magnitude $\|\Delta\boldsymbol{\epsilon}\|$ varies significantly across different editing tasks and architectures due to posterior variance shifts (Information Imbalance). Therefore, absolute thresholding is infeasible. We employ this z-score normalization to transform the raw gradients into a scale-invariant semantic signal. This allows us to use fixed relative thresholds ($\sigma$-levels) that generalize across prompts, seeds, and individual edits within a given baseline model. As a result, weak background signals are restored to a comparable scale, while strong object signals are prevented from overwhelming the map.

Based on empirical analysis (see Figure C.2), the extreme tails of this semantic map correspond to the cleanest semantic signals. Intermediate values often represent mixtures of object and background, making them unsuitable for disentangled edits. We therefore define two primary semantic layers using fixed thresholds that proved stable across models and prompts: a *background/style layer* ($M_{\text{sem}} < 0.6$) and an *object-core layer* ($M_{\text{sem}} \geq 3.0$).

## 4.2 STAGE 2: DISENTANGLED APPLICATION MODALITIES

The extracted semantic map $M_{\text{sem}}$ enables two distinct, training-free editing modalities.

**Static Mask Blending for Maximum Fidelity (Optional).** This static mask is optional and acts as a loose, permissive spatial constraint determined by the editing intent. Specifically, we define the active editing region using a coarse threshold: targeting **high-magnitude areas ($M_{\text{sem}} \geq 0.6$) for object edits**, and **low-magnitude areas ($M_{\text{sem}} < 0.6$) for background edits**. Unlike methods relying on strict hard boundaries, this mask is designed to be broad, preventing edits from drifting into completely irrelevant regions while leaving the semantic boundaries flexible. Only for tasks requiring strict identity preservation do we impose a tighter constraint by explicitly excluding high-magnitude object cores ($M_{\text{sem}} \geq 3.0$). At each step $t$, the edited latent $\mathbf{x}_{t-1}^{\text{pred}}$ is blended with the corresponding source latent $\mathbf{x}_{t-1}^{\text{src}}$, guaranteeing that unmasked regions remain unchanged:

$$\mathbf{x}_{t-1} \leftarrow \mathbf{x}_{t-1}^{\text{pred}} \odot M_{\text{final}} + \mathbf{x}_{t-1}^{\text{src}} \odot (1 - M_{\text{final}}), \tag{8}$$

where $M_{\text{final}}$ is obtained by thresholding $M_{\text{sem}}$ to get a coarse mask and refining it with morphological closing (see Appendix B).

**Dynamic Guidance Modulation (Default).** Our default modality offers greater flexibility by dynamically modulating guidance at each step. Although theoretically defined as a continuous map, **in practice, we binarize $W_{\text{sem},t}$ based on the z-score of the instantaneous $\|\Delta\epsilon_t\|$ (using $< 0.6\sigma$ for background edits and $\geq 3.0\sigma$ for object edits) to ensure stability and prevent boundary artifacts.** The guidance is then modulated element-wise:

$$\tilde{\epsilon}_\theta(x_t, c) \;=\; \epsilon_\theta(x_t, c_{\text{src}}) \;+\; \gamma \cdot \big(\Delta\epsilon_t \odot W_{\text{sem},t}\big). \tag{9}$$

This enables *region-specific guidance scaling*: background edits (low-information, high-variance) can be amplified with large $\gamma$ (e.g., 20–40) without destabilizing object regions (already high-information).

Importantly, this dynamic modulation is a direct operationalization of the information-field perspective from Section 3: by locally scaling weak, high-uncertainty regions while leaving strong, low-uncertainty regions untouched, we effectively re-balance the Fisher information disparity inherent in diffusion guidance.

**Notes on stability, scale, and hyperparameters.** Because the binarized $W_{\text{sem},t}$ strictly isolates the target region, background edits remain stable even under large local scales, as the amplification is explicitly prevented from bleeding into the object core. Static masking serves as an optional secondary safety filter, but dynamic modulation alone suffices for most edits and is our default. **Regarding hyperparameters, while specific thresholds vary per baseline architecture (e.g., to account for distinct noise schedules), they remain invariant across diverse datasets and prompts. Once set for a baseline, no per-image tuning is required.** The complete procedure is detailed in Algorithm 1 in the Appendix.

## 5 EXPERIMENTS

We conduct a comprehensive evaluation of **Prism-Edit** to validate our core claims: (1) the Semantic Scale Hypothesis is a general principle, and (2) our method enables state-of-the-art disentangled editing. Our evaluation spans multiple foundational models, including Stable Diffusion v1.5, v3, and `FLUX.1`, to demonstrate model-agnosticism.

**Implementation Details.** Unless otherwise specified, all experiments are performed using the default schedulers and step counts for each model. Per our theoretical motivation in Section 4, we apply a large region-specific guidance scale ($\gamma \in [20, 40]$) on low-magnitude regions for background edits, while conventional scales are used for object edits. Our ablation studies (see Appendix, Figures C.3 and C.4) confirm that this targeted amplification effectively modifies low-energy regions without introducing artifacts or destabilizing high-energy object structures. Hyperparameters for our static masking modalities are detailed in Table B.1. To ensure the reliability of our approach, we further verified that our Semantic Scale Hypothesis remains robust across different sampling conditions, including varying inversion techniques (e.g., DDIM vs. DPM-Solver Lu et al. (2022); Hong et al. (2024)) and target prompts, as detailed in **Appendix C.9**.

### 5.1 QUANTITATIVE EVALUATION

We evaluate on the standard **Wild-TI2I** and **ImageNet-R-TI2I** benchmarks. To specifically probe disentanglement, we partition Wild-TI2I into object-centric and background-centric subsets. We report standard metrics: DINOv2 (Oquab et al., 2024) for semantic alignment, SSIM (Wang et al., 2004) for structural preservation, and CLIP score (Radford et al., 2021) for text alignment.

**On the Trade-off between Disentanglement and Global Alignment.** While the CLIP score is a valuable metric for overall text-alignment, we observed that it does not always capture the nuances of disentangled editing. Since CLIP is known to bias towards global image modifications, baseline methods that alter the entire scene often achieve higher scores even when they fail to preserve identity. In contrast, Prism-Edit strictly preserves the unedited regions, which naturally limits this global drift. Consequently, while this may result in a slight CLIP decrease, it yields significantly higher semantic fidelity (DINO/SSIM), as intended. To provide a more complete picture, we introduce a supplementary metric:

$$\text{DINO/SSIM} \;=\; \frac{\text{DINOv2 (object similarity)}}{\text{SSIM (background preservation)}},$$

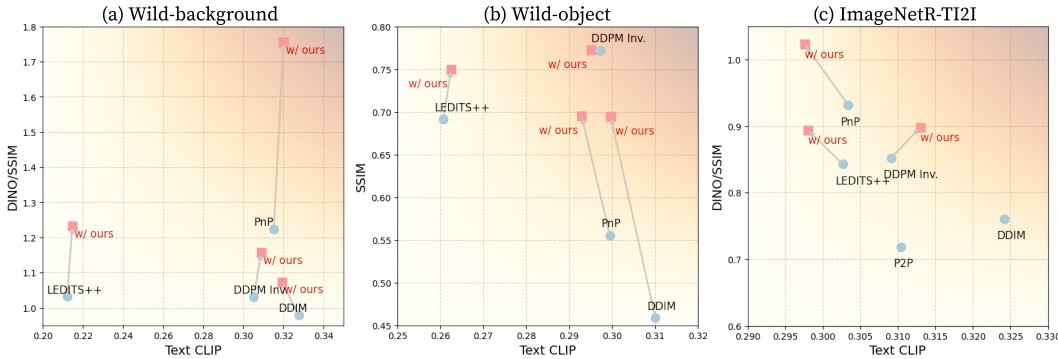

Figure 4: **Quantitative evaluation of Prism-Edit.** We report DINOv2, SSIM, and CLIP on Wild-TI2I (split into background/object subsets) and ImageNet-R-TI2I. Our method consistently improves DINOv2 similarity and maintains SSIM, validating disentangled editing performance.

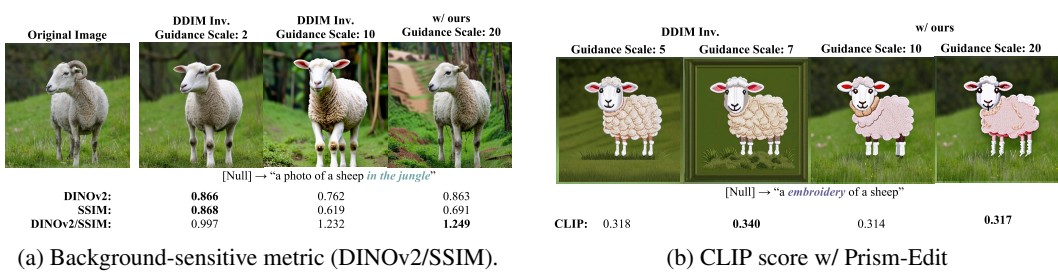

(a) Background-sensitive metric (DINOv2/SSIM).  (b) CLIP score w/ Prism-Edit

Figure 5: **Analysis of the Disentanglement Trade-off.** (a) Our method improves background-aware editing fidelity (DINOv2/SSIM). (b) This demonstrates that our edits prioritize disentanglement, which is not always captured by global text-alignment metrics like CLIP.

This ratio is designed to explicitly measure the success of preserving the primary object while altering the background. As shown in Figure 4, our method consistently outperforms baselines on this metric. This trade-off is further illustrated in Figure 5: while CLIP scores may plateau (Fig. 5b), our method maintains a high DINO/SSIM ratio (Fig. 5a), highlighting its effectiveness in disentangled editing.

## 5.2 QUALITATIVE EVALUATION

To validate the universality of our approach, we evaluate Prism-Edit's performance as a plug-and-play enhancement for established editing methods on Stable Diffusion v1.5. Detailed descriptions of these baselines and the integration methodology are provided in Appendix B.3. As shown in Figure 6, Prism-Edit consistently corrects common failure modes of baselines like DDIM/DDPM Inversion, PnP, and LEDITS++. For instance, when editing "an origami of a hummingbird" to "a sketch of a parrot," Prism-Edit successfully disentangles the object's identity ('parrot') from its style ('sketch'), a task where baselines often fail. This demonstrates the broad utility of our principled guidance decomposition. Further results in Figure 7 confirm our method's model-agnostic performance. Furthermore, we demonstrate that our Semantic Scale Hypothesis generalizes beyond object-centric images. Our analysis on object-scarce scenes (e.g., landscapes, textures) confirms that the guidance magnitude effectively disentangles implicit local structures from global atmosphere, as detailed in Appendix C.8. We demonstrate robust background and object edits on modern architectures like Stable Diffusion v3, and showcase Prism-Edit's utility on `FLUX.1` by integrating it as a plug-and-play enhancement for existing editors like RF-Inversion (Rout et al., 2025) and Stable-flow (Avrahami et al., 2025).

## 5.3 CAUSAL VALIDATION OF SEMANTIC DISENTANGLEMENT

A key prediction of our hypothesis is that distinct semantic layers can be edited independently. To provide causal evidence, we design prompts that require simultaneous object and background changes.

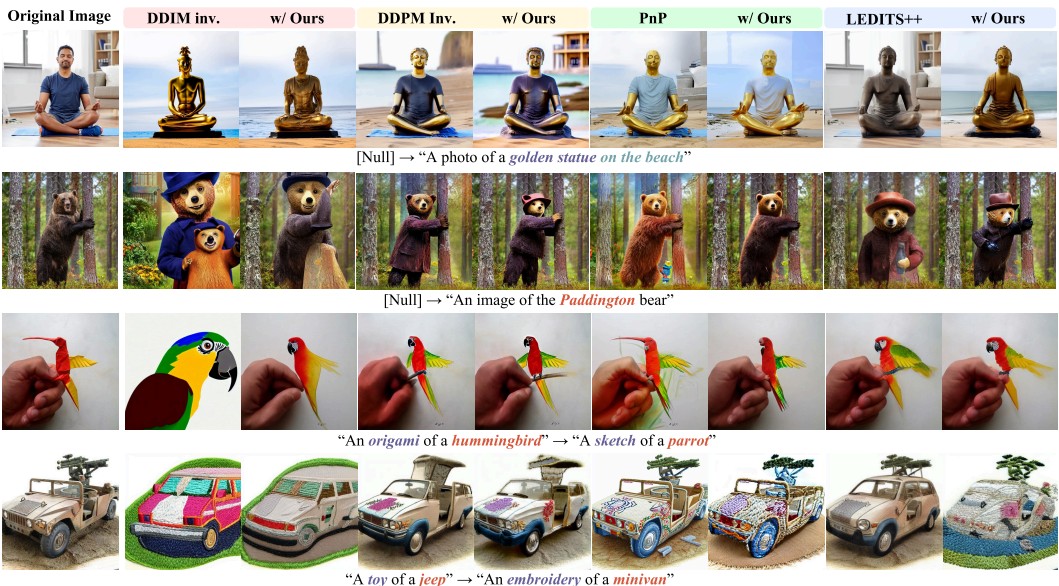

Figure 6: **Prism-Edit as a Universal Enhancement Module.** Our method, integrated with various editing techniques on SD v1.5, consistently corrects common failure modes like semantic leakage (rows 3-4) and incomplete edits (rows 1-2).

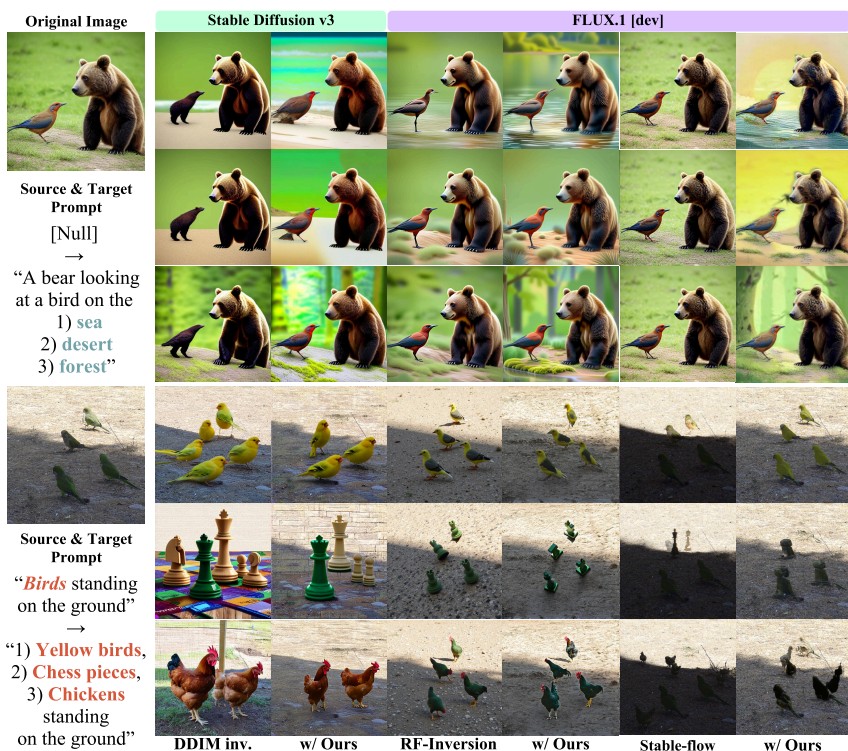

Figure 7: **Model-agnostic editing.** Results on SD v3 and `FLUX.1`. Prism-Edit enables faithful background modifications (rows 1–2) and robust object edits (rows 3–4).

We then apply Prism-Edit in two controlled settings: (i) editing only high-magnitude (object) signals, and (ii) editing only low-magnitude (background) signals. As shown in Figure 8, the results are cleanly disentangled. Modifying high-magnitude signals alters the object's identity while preserving

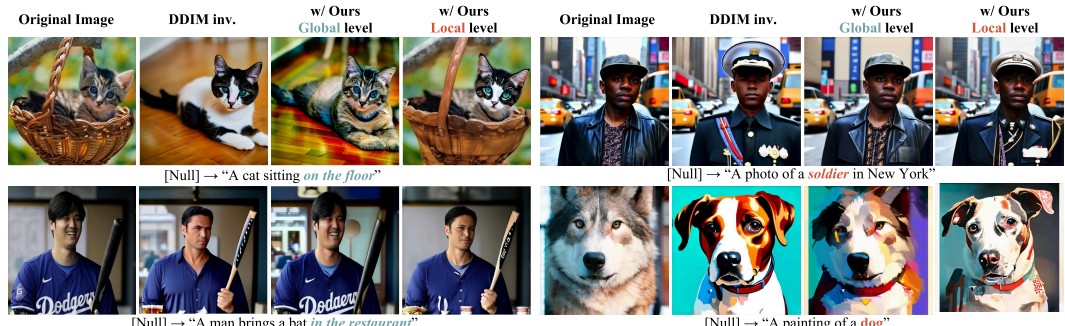

Figure 8: **Semantic layer disentanglement.** The results show a clear causal separation between **Local level** (high-magnitude) edits that alter object identity, and **Global level** (low-magnitude) edits that alter background and style.

the background, and vice-versa. This experiment directly validates that guidance magnitude causally corresponds to the semantic scales we identified.

# 6    LIMITATIONS

Prism-Edit has several limitations. Our theoretical analysis assumes a Gaussian posterior, which simplifies derivations but does not perfectly reflect the true diffusion process. The framework also requires manual specification of editing intent and relies on fixed z-score thresholds to separate semantic layers, introducing both user intervention and heuristic design choices. In addition, the effectiveness of our method is influenced by the baseline diffusion model into which it is plugged, meaning that gains may vary depending on the underlying architecture.

# 7    CONCLUSION

We introduced the **Semantic Scale Hypothesis**, framing guidance magnitude ($\|\Delta\epsilon\|$) as an information-theoretic signal that reflects a semantic hierarchy. Based on this principle, our training-free method **Prism-Edit** adaptively decomposes the guidance field to enable more disentangled edits, particularly in challenging background regions. Rather than relying solely on spatial masks, this perspective highlights the role of signal-level structure within diffusion guidance. Future work may explore automatic detection of user intent and more adaptive layer selection, moving toward a practical zero-shot editing pipeline.

## ACKNOWLEDGMENTS

This work was supported in part by the Institute of Information & Communications Technology Planning & Evaluation (IITP) grants funded by the Korea government (MSIT), including Artificial Intelligence Graduate School Program (No. RS-2019-II190079, Korea University), (Part 2) Few-Shot Learning of Causal Inference in Vision and Language for Decision Making project (No. RS-2022-II220959), and National AI Research Lab Project (No. RS-2024-00457882).

## ETHICS STATEMENT

The authors adhere to the ICLR Code of Ethics. Our work introduces a method for text-guided image editing. We acknowledge that, like all generative models, this technology could potentially be misused for creating misleading or harmful content. However, the primary focus of our research is to provide a deeper understanding of the internal mechanisms of diffusion models and to offer controllable tools for creative and research purposes. We believe that by making the underlying principles of these models more transparent and controllable, our work contributes to a more responsible development path for generative AI.

## REPRODUCIBILITY STATEMENT

We are committed to ensuring the reproducibility of our research. To this end, our source code is included in the supplementary material and will be made publicly available upon publication. The appendix provides comprehensive details for replication: the full algorithm for Prism-Edit is presented in Algorithm 1, all hyperparameters for our experiments and baselines are listed in Table B.1, and the theoretical derivation of the Semantic Scale Hypothesis is available in Appendix A. The appendix also contains further details on our experimental setup, including additional results.

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

## A   DERIVATION OF THE SEMANTIC SCALE HYPOTHESIS

We provide here a more detailed derivation linking the magnitude of the guidance difference vector $\Delta\epsilon$ to posterior variance and information density, as introduced in Section 3.

**Step 1: Connection between Posterior Mean and Score Function via Tweedie's Formula.**   For a noisy observation $\mathbf{x}_t$ obtained by corrupting a clean image $\mathbf{x}_0$ with Gaussian noise of variance $\sigma_t^2$, i.e., $\mathbf{x}_t \sim q(\mathbf{x}_t|\mathbf{x}_0)$, Tweedie's formula (Efron, 2011) connects the posterior mean of the clean image to the score of the noisy data distribution:

$$\mathbb{E}[\mathbf{x}_0 \mid \mathbf{x}_t, c] = \mathbf{x}_t + \sigma_t^2 \nabla_{\mathbf{x}_t} \log p(\mathbf{x}_t \mid c). \tag{A.1}$$

**Step 2: Guidance Difference as a Difference of Scores.**   In standard diffusion models with $\epsilon$-parameterization, the noise predictor $\epsilon_\theta$ is trained to approximate the scaled score function: $\epsilon_\theta(\mathbf{x}_t, c) \propto -\sigma_t \nabla_{\mathbf{x}_t} \log p(\mathbf{x}_t \mid c)$. Therefore, the guidance difference vector $\Delta\epsilon$ between a source condition $c_1$ and a target condition $c_2$ is proportional to the difference in their respective score functions:

$$\Delta\boldsymbol{\epsilon}(\mathbf{x}_t; c_1, c_2) \propto \sigma_t \left( \nabla_{\mathbf{x}_t} \log p(\mathbf{x}_t \mid c_1) - \nabla_{\mathbf{x}_t} \log p(\mathbf{x}_t \mid c_2) \right). \tag{A.2}$$

**Step 3: Relating Guidance Magnitude to Posterior Mean Shift.**   By rearranging Eq. A.1 and combining it with Eq. A.2, we establish a direct relationship between the magnitude of the guidance vector and the shift in the posterior mean estimate:

$$\|\Delta\boldsymbol{\epsilon}(\mathbf{x}_t; c_1, c_2)\| \; \propto \; \frac{\|\Delta\boldsymbol{\mu}_t\|}{\sigma_t}, \quad \text{where} \quad \Delta\boldsymbol{\mu}_t := \mathbb{E}[\mathbf{x}_0 \mid \mathbf{x}_t, c_2] - \mathbb{E}[\mathbf{x}_0 \mid \mathbf{x}_t, c_1]. \tag{A.3}$$

This shows that a large guidance magnitude corresponds to a large shift in the model's estimate of the clean image when the condition changes from $c_1$ to $c_2$.

**Step 4: The Role of Posterior Variance and Information Density.**   The magnitude of the posterior shift $\Delta\boldsymbol{\mu}_t$ is determined by the "sharpness" or certainty of the posterior distribution $p(\mathbf{x}_0 \mid \mathbf{x}_t, c)$.

- **Low-Variance Posteriors (High Information Density):** In image regions with rich structure and detail (e.g., objects), the posterior distribution is sharply peaked (low variance). Here, the model is highly certain about the content. A change in condition $(c_1 \to c_2)$ forces a significant shift in this sharp distribution, leading to a large $\Delta\boldsymbol{\mu}_t$ and thus a large-magnitude $\|\Delta\boldsymbol{\epsilon}\|$.
- **High-Variance Posteriors (Low Information Density):** In smooth, less structured regions (e.g., backgrounds), the posterior is diffuse and spread out (high variance). The model is uncertain about the precise content. The same conditional change results in a smaller adjustment to the broad distribution, yielding a small $\Delta\boldsymbol{\mu}_t$ and consequently a small-magnitude $\|\Delta\boldsymbol{\epsilon}\|$.

**Conclusion.**   The Semantic Scale Hypothesis is a direct consequence of this relationship. The guidance magnitude $\|\Delta\boldsymbol{\epsilon}\|$ acts as a proxy for the information-theoretic sharpness of the posterior. Structured objects correspond to high-information regions and yield large-magnitude guidance, while smooth backgrounds correspond to low-information regions and yield small-magnitude guidance. This theoretical relationship is formalized in Theorem 1, which provides closed-form bounds on the guidance magnitude. A sketch of the proof follows:

**Proof sketch of Theorem 1.**   Start from Tweedie's relation (App. A) implying $\|\Delta\boldsymbol{\epsilon}\| \propto \|\Delta\boldsymbol{\mu}_t\|/\sigma_t$. Under the Gaussian approximation,

$$(\Delta\boldsymbol{\mu}_t)^\top \boldsymbol{\Sigma}_{c_2}^{-1} \Delta\boldsymbol{\mu}_t \; = \; 2\, D_{\mathrm{KL}}(\mathcal{N}(\boldsymbol{\mu}_{c_1}, \boldsymbol{\Sigma}_{c_1}) \,\|\, \mathcal{N}(\boldsymbol{\mu}_{c_2}, \boldsymbol{\Sigma}_{c_2})) - \Psi(\boldsymbol{\Sigma}_{c_1}, \boldsymbol{\Sigma}_{c_2}),$$

by rearranging the closed-form Gaussian KL. Using the Rayleigh quotient,

$$\lambda_{\min}(\boldsymbol{\Sigma}_{c_2}^{-1}) \, \|\Delta\boldsymbol{\mu}_t\|^2 \; \leq \; (\Delta\boldsymbol{\mu}_t)^\top \boldsymbol{\Sigma}_{c_2}^{-1} \Delta\boldsymbol{\mu}_t \; \leq \; \lambda_{\max}(\boldsymbol{\Sigma}_{c_2}^{-1}) \, \|\Delta\boldsymbol{\mu}_t\|^2,$$

i.e.,

$$\frac{\|\Delta\boldsymbol{\mu}_t\|^2}{\lambda_{\max}(\boldsymbol{\Sigma}_{c_2})} \; \leq \; (\Delta\boldsymbol{\mu}_t)^\top \boldsymbol{\Sigma}_{c_2}^{-1} \Delta\boldsymbol{\mu}_t \; \leq \; \frac{\|\Delta\boldsymbol{\mu}_t\|^2}{\lambda_{\min}(\boldsymbol{\Sigma}_{c_2})}.$$

Combining with the KL identity and rescaling by $\sigma_t^{-2}$ yields the upper/lower bounds in Eqs. equation 4–equation 6. The symmetric versions follow by swapping $(c_1, c_2)$. $\qquad\square$

# B  IMPLEMENTATION AND HYPERPARAMETERS

## B.1  ALGORITHM DETAILS

The complete algorithm for the Prism-Edit framework is detailed in **Algorithm 1**. It outlines the two-stage process for semantic map extraction and disentangled application. **Algorithm 2** specifies the morphological closing operation used to refine the binary mask, ensuring spatial contiguity.

---

**Algorithm 1** Prism-Edit (Full Version with Optional Static Mask Refinement)

---

**Require:** Source prompt $c_{\text{src}}$, target prompt $c_{\text{tgt}}$, probe interval $\{t_{900}, \ldots, t_{800}\}$
1: Initialize latent $\mathbf{x}_T \sim \mathcal{N}(0, I)$
2: **// Stage 1: Semantic Map Extraction**
3: **for** $t_i$ in probe interval **do**
4:     $\Delta\boldsymbol{\epsilon}_{t_i} \leftarrow \epsilon_\theta(\mathbf{x}_{t_i}, c_{\text{tgt}}) - \epsilon_\theta(\mathbf{x}_{t_i}, c_{\text{src}})$
5: **end for**
6: $M_{\text{sem}} \leftarrow z\text{-score}\left(\frac{1}{N}\sum_i \|\Delta\boldsymbol{\epsilon}_{t_i}\|\right)$
7: **// Optional: Static Mask Generation**
8: **if** target is object **then**
9:     $M_{\text{base}} \leftarrow (M_{\text{sem}} \geq 0.6)$
10: **else**                                             ▷ target is background
11:     $M_{\text{base}} \leftarrow (M_{\text{sem}} < 0.6)$
12: **end if**
13: $M_{\text{filled}} \leftarrow \text{Mask-Refinement}(M_{\text{base}})$                    ▷ See Algorithm 2
14: **if** identity preservation mode **then**
15:     $M_{\text{exclude}} \leftarrow (M_{\text{sem}} \geq 3.0)$
16:     $M_{\text{final}} \leftarrow \text{clamp}(M_{\text{filled}} - M_{\text{exclude}}, 0, 1)$
17: **else**
18:     $M_{\text{final}} \leftarrow M_{\text{filled}}$
19: **end if**
20: **for** $t = T, \ldots, 1$ **do**                    ▷ **Stage 2: Disentangled Application**
21:     **// Dynamic guidance modulation (always on)**
22:     $\Delta\boldsymbol{\epsilon}_t \leftarrow \epsilon_\theta(\mathbf{x}_t, c_{\text{tgt}}) - \epsilon_\theta(\mathbf{x}_t, c_{\text{src}})$
23:     **// Binarize based on editing intent (e.g., $\geq 3.0\sigma$ for object, $< 0.6\sigma$ for bg)**
24:     $W_{\text{sem},t} \leftarrow \text{Binarize}\left(z\text{-score}(\|\Delta\boldsymbol{\epsilon}_t\|) \text{ meets } \tau\right)$
25:     $\tilde{\epsilon}_\theta \leftarrow \epsilon_\theta(\mathbf{x}_t, c_{\text{src}}) + \gamma \cdot (\Delta\boldsymbol{\epsilon}_t \odot W_{\text{sem},t})$
26:     $\mathbf{x}_{t-1}^{\text{pred}} \leftarrow \mathcal{S}(\mathbf{x}_t, \tilde{\epsilon}_\theta, t)$
27:     **// Static blending (optional)**
28:     **if** static mask mode **then**
29:         $\mathbf{x}_{t-1} \leftarrow \mathbf{x}_{t-1}^{\text{pred}} \odot M_{\text{final}} + \mathbf{x}_{t-1}^{\text{src}} \odot (1 - M_{\text{final}})$
30:     **else**
31:         $\mathbf{x}_{t-1} \leftarrow \mathbf{x}_{t-1}^{\text{pred}}$
32:     **end if**
33: **end for**
34: **return** Edited image $\hat{\mathbf{x}}_0$

---

---

**Algorithm 2** Mask Refinement (Morphological Closing)

---

**Require:** Base mask $M_{\text{base}} \in \{0, 1\}^{H \times W}$, number of iterations $K$
   *We apply morphological closing (dilation followed by erosion) to ensure the semantic mask is contiguous and free of small holes.*
1: $M \leftarrow M_{\text{base}}$
2: **for** $k = 1$ to $K$ **do**
3:     $M \leftarrow \text{Dilate}(M)$
4:     $M \leftarrow \text{Erode}(M)$
5: **end for**
6: **return** $M_{\text{filled}} \leftarrow M$

---

## B.2 EXPERIMENTAL SETUP AND HYPERPARAMETERS

All experiments employ a null-text inversion approach to prioritize content preservation. In Stable Diffusion v1.4/v1.5 (Rombach et al., 2022), we use 100 timesteps for DDIM inversion (Song et al., 2021) with a guidance scale of 1, and 50 steps for editing. Table B.1 lists the specific hyperparameters used to enhance various baseline methods.

Table B.1: Hyperparameter settings for applying Prism-Edit to various baseline methods.

| Baseline Method | Inv. Steps | Edit Steps | Base Str. | $\gamma_{\mathbf{obj}}$ | $\gamma_{\mathbf{bg}}$ | Thresholds (Obj / Bg) |
|---|---|---|---|---|---|---|
| DDIM inv. | 100 | 100 | 7.5 | 7.5 | 30 | $\geq 3.0 \, / < 0.4$ |
| P2P | 100 | 100 | 9 | 15 | 30 | $\geq 3.0 \, / < 1.0$ |
| PnP | 1000 | 50 | 10 | 25 | 40 | $\geq 3.0 \, / < 2.0$ |
| DDPM inv. | 100 | 100 | 15 | 25 | 40 | $\geq 2.0 \, / < 1.0$ |
| LEDITS++ | 50 | 50 | 10 | 20 | 30 | $\geq 3.0 \, / < 0.6$ |

## B.3 BASELINE INTEGRATION DETAILS

To demonstrate the model-agnostic nature of Prism-Edit, we integrated it with several state-of-the-art editing methods. Since Prism-Edit operates directly on the guidance vector $\Delta\epsilon$, it can be seamlessly combined with methods that manipulate internal network features or attention maps.

- **Prompt-to-Prompt (P2P) (Hertz et al., 2023) & Plug-and-Play (PnP) (Tumanyan et al., 2023):** Both methods primarily control image structure by manipulating cross-attention maps or injecting spatial features during the denoising process.

  **Integration:** We integrate Prism-Edit by strictly respecting the original pipeline for attention/feature injection. However, at the guidance computation stage of each sampling step, we replace the standard classifier-free guidance vector with our proposed Prism-Edit modulation (**specifically, using the Dynamic Guidance Modulation mode only**). This allows us to combine the structural stability of P2P/PnP with Prism-Edit's semantic disentanglement capability, enabling selective amplification or preservation of specific regions without relying on hard masking.

- **LEDITS++ (Brack et al., 2024):** The original LEDITS++ method employs a native mechanism that identifies editing regions by thresholding the difference in guidance vectors via quantiles (similar to DiffEdit's masking strategy).

  **Integration:** We explicitly **bypass** this native quantile-based masking step. Instead, consistent with our integration for P2P and PnP, we apply Prism-Edit by directly modulating the classifier-free guidance term (**using Dynamic Guidance Modulation only**) during the sampling process. This demonstrates that our gradient-based modulation offers a more robust alternative to the hard-thresholding masks originally employed by LEDITS++.

In all cases, Prism-Edit does not require retraining or modifying the internal architecture of the baselines, confirming its plug-and-play capability.

## C ADDITIONAL EXPERIMENTAL RESULTS AND ANALYSES

### C.1 GLOBAL VS. LOCAL CHANGES

To analyze the effectiveness of our proposed method on both global and local image modifications, we consider two representative image-to-image translation scenarios: transitioning a Yosemite landscape from summer to winter (a global change) and transforming a horse into a zebra (a local change). We utilize the same Yosemite (summer↔winter) and Horse (horse↔zebra) datasets as employed in the CycleGAN (Zhu et al., 2017) benchmark, enabling direct comparison with existing methods. Experiments were conducted with both null-text and valid-text prompts. For valid-text prompts, we used "a photo of Yosemite in summer" → "A photo of Yosemite in winter" and "A photo of a horse" → "A photo of a zebra". For null-text prompts, the source prompt was null ("").

| Inversion | Guidance | Summer → Winter | | | | Horse → Zebra | | | |
|---|---|---|---|---|---|---|---|---|---|
| | | LPIPS (×100) ↓ | | CLIP$_{text}$ ↑ | | LPIPS (×100) ↓ | | CLIP$_{text}$ ↑ | |
| | | Null | Valid | Null | Valid | Null | Valid | Null | Valid |
| DDIM Inv. | CFG | 58.99 | 62.76 | **23.15** | **25.77** | 62.30 | 72.50 | 20.43 | 30.00 |
| | + Prism-Edit (Ours) | **36.75** | **28.39** | 21.53 | 22.18 | **30.94** | **56.51** | **21.26** | **30.65** |
| DDPM Inv. | CFG | 33.50 | 39.56 | 21.26 | **22.03** | 29.77 | 29.65 | **28.54** | **28.55** |
| | + Prism-Edit (Ours) | **33.42** | **28.52** | **21.59** | 21.29 | **25.92** | **27.66** | 28.44 | 28.54 |

Table C.1: **Quantitative Comparison of Image-to-Image Translation Results.** Comparison of our method with existing methods using null-text and valid-text sampling approaches, evaluated by LPIPS and CLIP scores to assess perceptual similarity and alignment with text prompts.

Table C.1 and Figure C.1 present the quantitative and qualitative findings. In all cases, our proposed method preserved structural information significantly better than the baselines, as evidenced by the lower LPIPS (Zhang et al., 2018) scores. While CLIP (Radford et al., 2021) scores were sometimes similar or slightly lower, this typically occurred when the baseline methods failed to maintain the original structure, causing a large deviation from the source image.

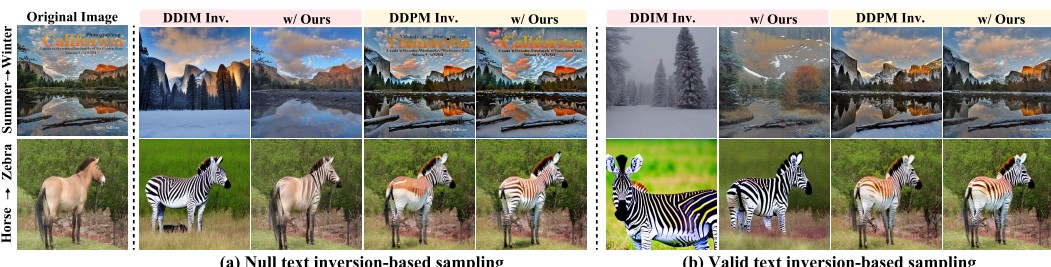

Figure C.1: **Qualitative comparison for global and local changes on CycleGAN datasets.** Our method (w/ Ours) is applied to two I2I translation tasks: global style change (Summer→Winter) and local object change (Horse→Zebra). Compared to baselines, our method better preserves the structural integrity of the source image (e.g., mountain layout, horse's pose) while successfully applying the target transformation.

## C.2 RATIONALE FOR SEMANTIC LAYER SELECTION

As discussed in Section 4, our framework is designed to operate on the tails of the semantic map's distribution. Figure C.2 provides the empirical validation for this design choice. We conduct an experiment to visualize which image content is affected by edits restricted to different intervals of the $M_{sem}$ map. The results clearly indicate that the intermediate intervals (e.g., $0.5 \leq \sigma \leq 2.0$) contain a mixture of object, background, and texture information. Edits applied to these regions often result in undesirable artifacts and semantic leakage. In contrast, the extreme low- and high-magnitude regions (tails) correspond to purer signals for background/style and object-core structure, respectively. Therefore, by selectively targeting these tails, Prism-Edit achieves cleaner, more disentangled manipulation.

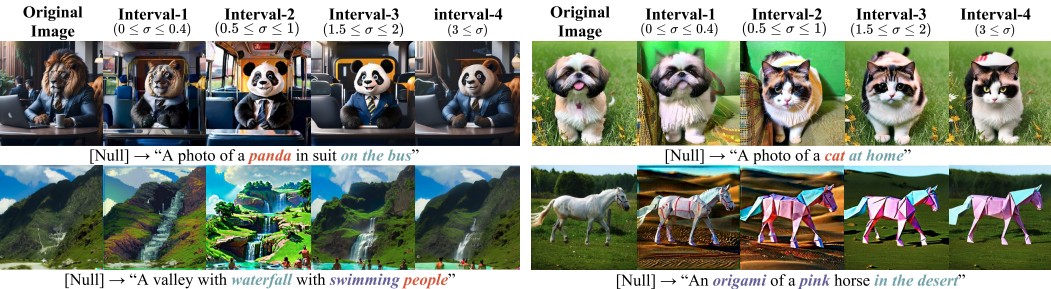

Figure C.2: **Impact of varying standard deviation thresholds for layer selection.** Edits are applied only to pixels within the specified $\sigma$-interval of the semantic map. The intermediate intervals (2 and 3) show a clear mixture of semantics, validating our design choice to operate on the tails (intervals 1 and 4) for disentangled editing.

## C.3 ABLATION STUDIES ON GUIDANCE SCALE $\gamma$

To validate our use of large, region-specific guidance scales ($\gamma$), we perform ablation studies on **two distinct editing modes** based on the targeted region of the semantic map ($M_{\text{sem}}$):

- **Outer-Interval Editing (Local Level):** This mode targets the tails of the distribution (e.g., $|M_{\text{sem}}| \geq k_{\text{outer}}$), corresponding to the highest- and lowest-magnitude signals. As these signals cleanly map to object cores and uniform backgrounds, this mode is primarily used for precise **object editing**.

- **Inner-Interval Editing (Global Level):** This mode targets the central part of the distribution (e.g., $|M_{\text{sem}}| < k_{\text{inner}}$), which contains the low-energy signals associated with overall style and texture. This mode is used for global **background and stylistic changes**.

Figures C.3 and C.4 show that Prism-Edit remains stable even with large $\gamma$ values in both modes. Figure C.3 shows that increasing $\gamma$ in Outer-Interval mode correctly strengthens the target object concept without corrupting the background. Conversely, Figure C.4 demonstrates that large $\gamma$ values (up to 200) in Inner-Interval mode can achieve dramatic stylistic changes while the high-energy object regions remain structurally intact, protected by our dynamic modulation.

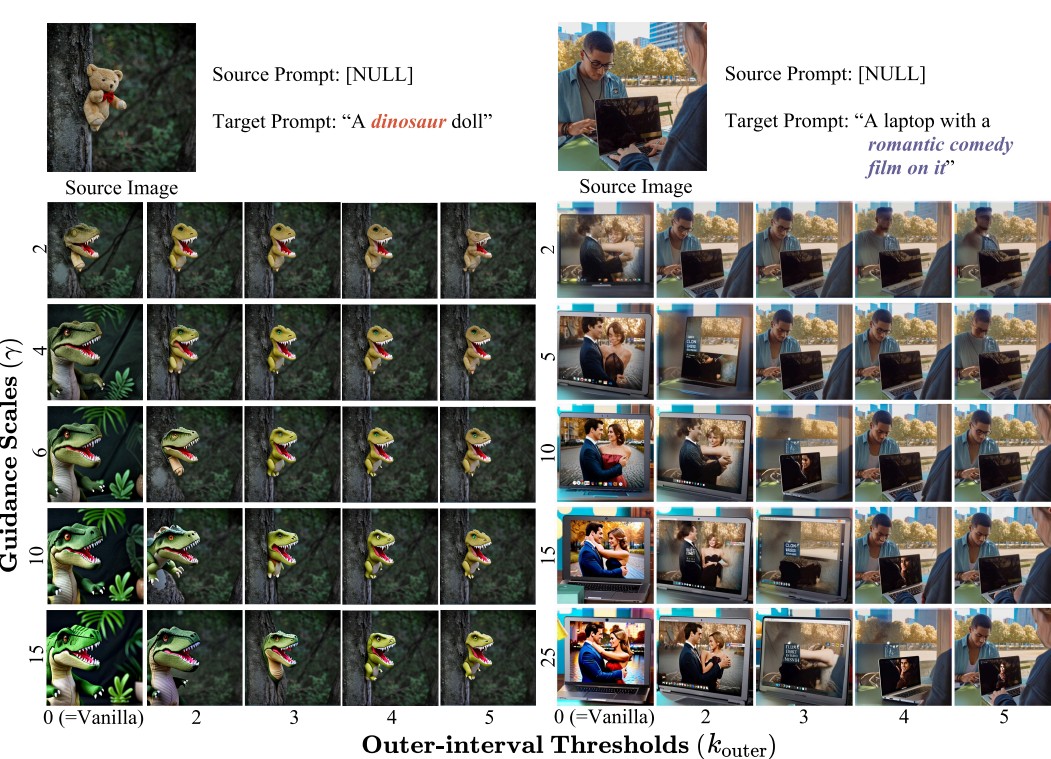

Figure C.3: **Ablation on guidance scale $\gamma$ for Outer-Interval (Object) editing.** The left example demonstrates object replacement, while the right showcases object insertion.

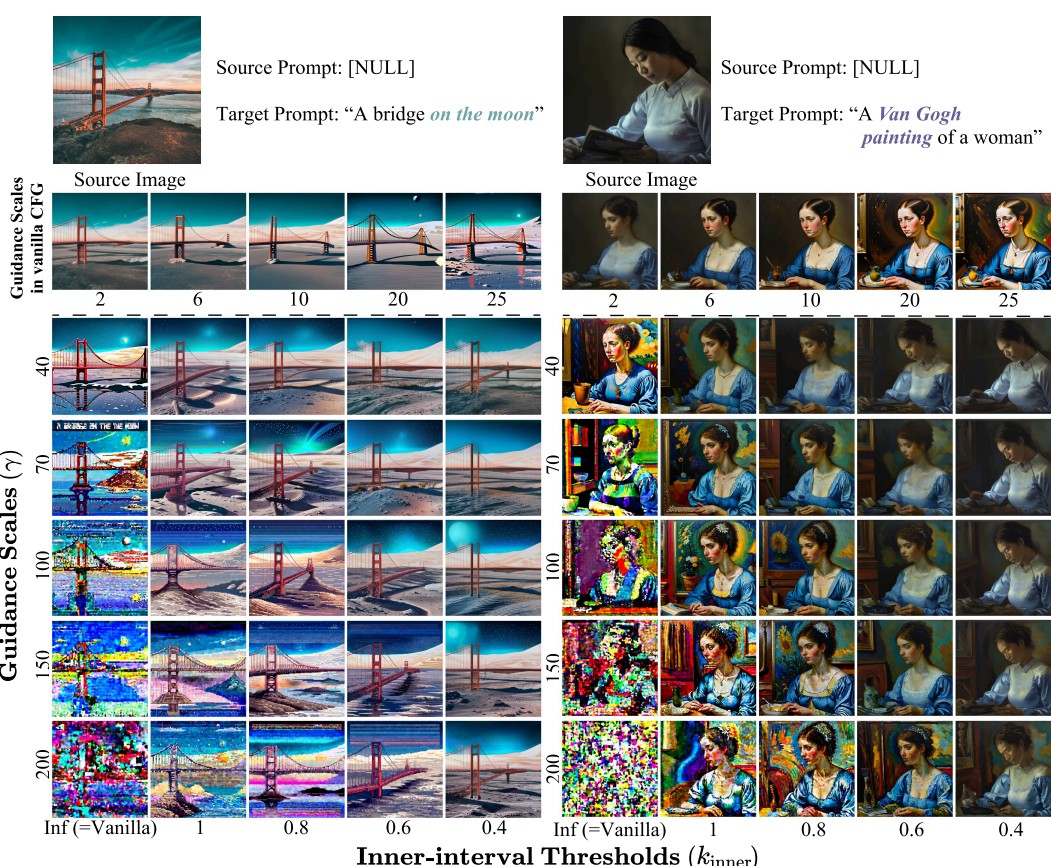

Figure C.4: **Ablation on guidance scale γ for Inner-Interval (Background/Style) editing.** The left example demonstrates background replacement, while the right showcases texture replacement.

## C.4    The Role of Negative Prompts

**Implementation Strategy.** During the inversion phase, we set the negative prompt to an empty string ("") to maximize reconstruction fidelity. During the editing (sampling) phase, we utilize the source prompt ($p_{\text{src}}$) as the negative prompt. This strategy effectively neutralizes the semantic features of the original concept, preventing them from leaking into the edited result.

Our framework is fully compatible with negative prompts. As shown in Figure C.5, negative prompts are crucial for overcoming the model's prior and achieving clean object replacement or attribute editing. However, as analyzed in Figure C.6, we found that for background-only edits, negative prompts can sometimes introduce subtle, undesirable changes to the foreground object. Therefore, we recommend using negative prompts primarily for object-focused manipulations.

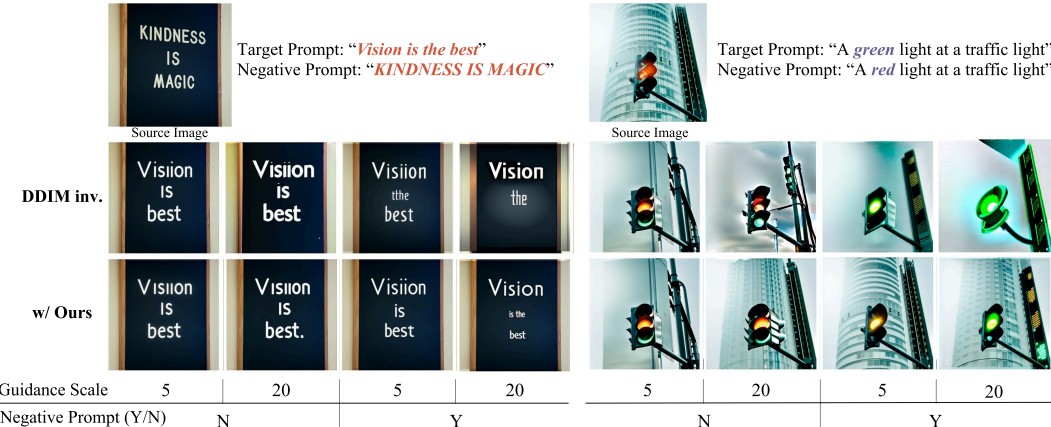

Figure C.5: **Effectiveness of negative prompts with Prism-Edit.** Negative prompts are essential for clean text replacement (left) and precise attribute editing (right).

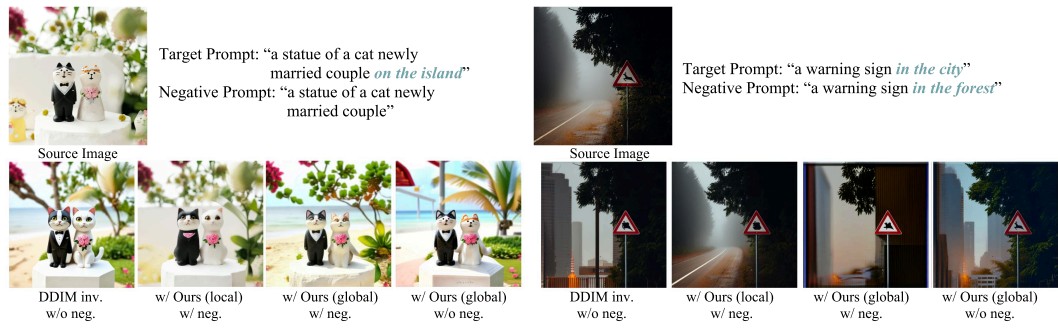

Figure C.6: **Interaction between negative prompts and background editing.** When modifying only the background, adding a negative prompt can cause minor semantic leakage into the foreground object (rightmost two images).

## C.5    Additional Qualitative and Quantitative Results

We provide further qualitative and quantitative results in Tables C.2-C.4 and Figures C.7-C.9.

Table C.2: Wild-background results.

| Method | PnP | DDIM | DDPM Inv. | LEDITS++ |
|---|---|---|---|---|
| **CLIP↑** | | | | |
| original | 0.3151 | **0.3275** | 0.3051 | 0.2120 |
| + ours | **0.3200** | 0.3192 | **0.3088** | **0.2147** |
| **DINO/SSIM ↑** | | | | |
| original | 1.2240 | 0.9796 | 1.0310 | 1.0317 |
| + ours | **1.7555** | **1.0731** | **1.1571** | **1.2328** |

Table C.3: Wild-object results.

| Method | PnP | DDIM | DDPM inv. | LEDITS++ |
|---|---|---|---|---|
| **CLIP↑** | | | | |
| original | **0.2995** | **0.3101** | **0.2972** | 0.2607 |
| + ours | 0.2929 | 0.2997 | 0.2901 | **0.2625** |
| **SSIM↑** | | | | |
| original | 0.5554 | 0.4588 | 0.7721 | 0.6914 |
| + ours | **0.6952** | **0.6942** | **0.7722** | **0.7499** |

Table C.4: Imagenet-R-TI2I results.

| Method | PnP | DDIM | DDPM inv. | LEDITS++ | P2P |
|---|---|---|---|---|---|
| **CLIP↑** | | | | | |
| original | **0.3034** | 0.3242 | **0.3091** | **0.3027** | **0.3104** |
| + ours | 0.2976 | 0.3199 | 0.3130 | 0.2980 | 0.3087 |
| **DINO/SSIM↑** | | | | | |
| original | 0.9312 | 0.7604 | 0.8523 | 0.8426 | 0.7187 |
| + ours | **1.0228** | **0.8979** | **0.8939** | **0.9982** | **0.8812** |

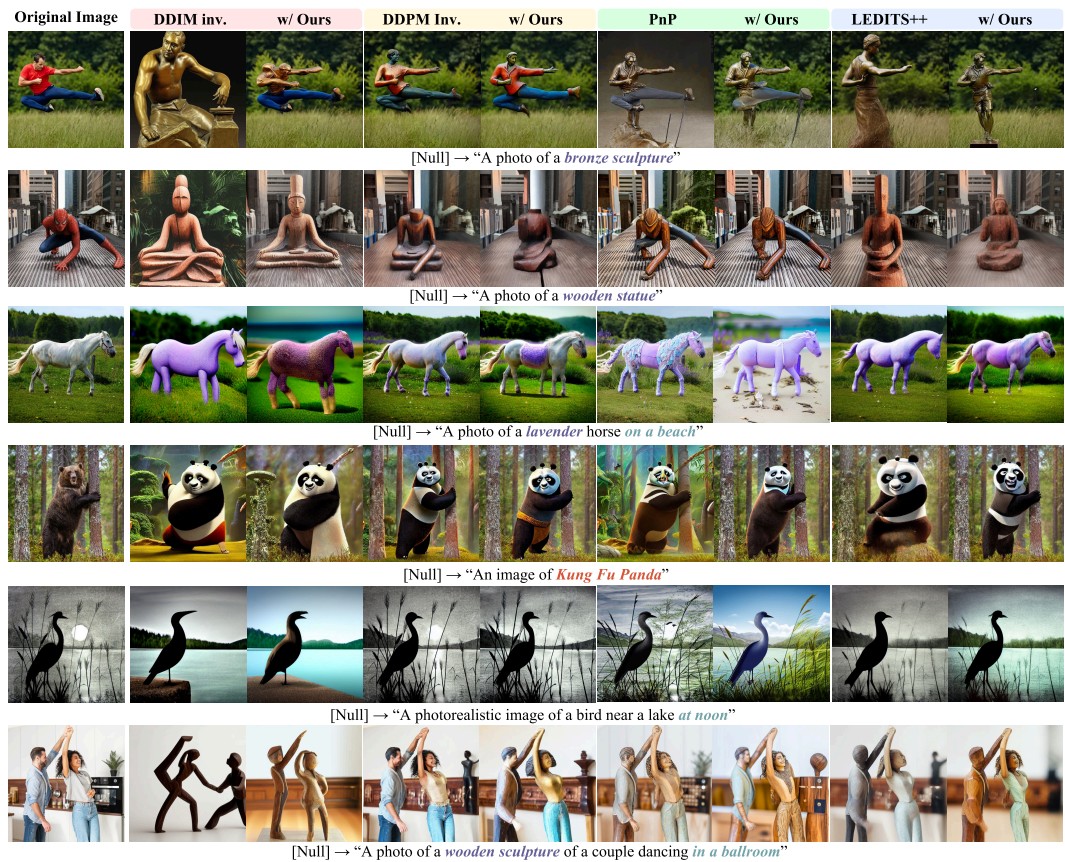

Figure C.7: **Further qualitative comparisons using null-text inversion.**

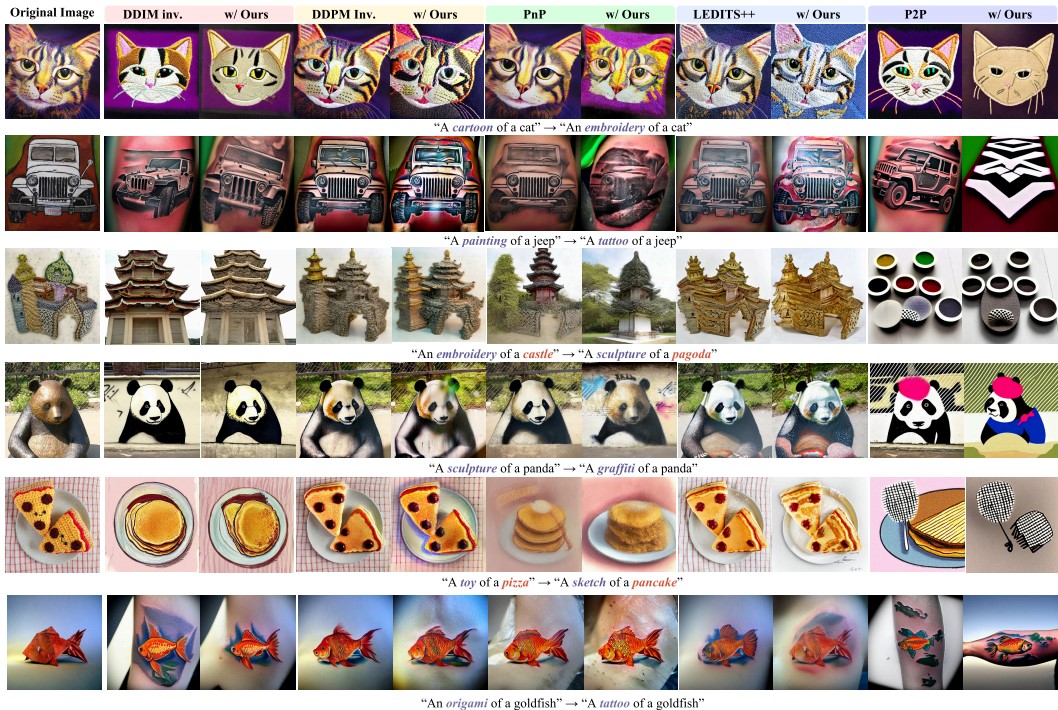

Figure C.8: **Further qualitative comparisons using valid-text inversion.**

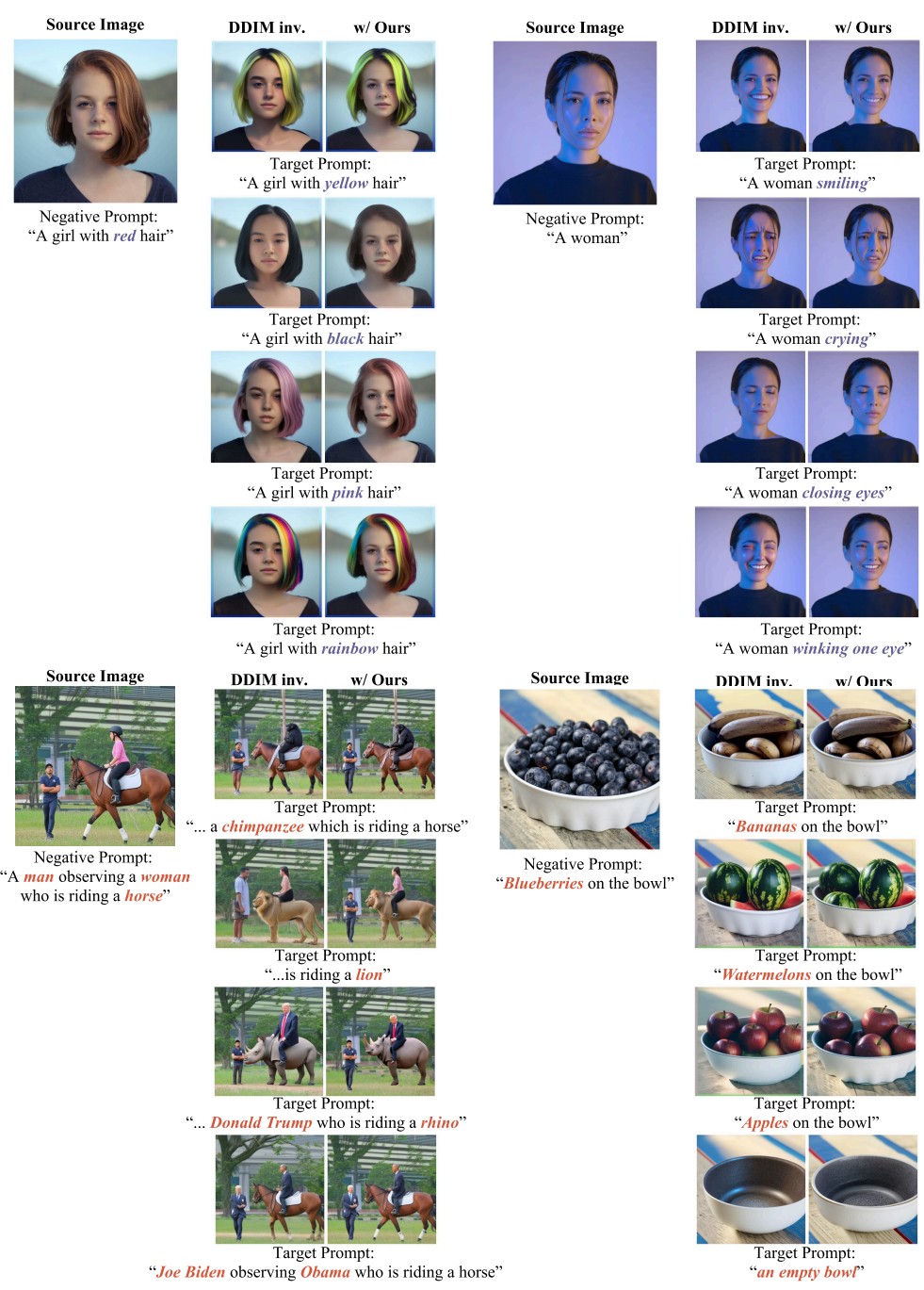

Figure C.9: **Additional attribute and object editing results with negative prompts.**

C.6 ABLATION STUDY ON PROBE INTERVAL SELECTION

To determine the optimal timestep window for semantic map extraction, we conducted a quantitative ablation study using the COCO 2017 validation set (Lin et al., 2014). We focused on the 'Person' class (50 randomly selected images, seed=0), as human subjects represent highly deformable objects that require both robust localization and structural flexibility during editing.

We evaluated the quality of the extracted semantic map $M_{\text{sem}}$ across different timestep intervals using two metrics:

- **Coverage (Recall):** Measures how well the semantic map covers the ground-truth object mask. High coverage indicates the map successfully captures the "semantic whole" of the object.
- **IoU (Intersection over Union):** Measures the spatial tightness of the map against the ground truth.

Table C.5: **Quantitative Analysis of Probe Intervals.** The interval $[900, 800]$ achieves the highest semantic coverage (0.9619), indicating it best captures the global object structure. Later steps (e.g., 500–400) show higher IoU but represent over-constraint to fine details.

| Timestep Window | Coverage (Recall) ↑ | Avg. IoU |
|---|---|---|
| 940–920 | 0.8839 | 0.2229 |
| 920–900 | 0.9206 | 0.2274 |
| **900–800 (Ours)** | **0.9619** | 0.2651 |
| 900–880 | 0.9109 | 0.2312 |
| 880–860 | 0.8818 | 0.2354 |
| 860–840 | 0.9254 | 0.2592 |
| 800–780 | 0.9026 | 0.2762 |
| 800–700 | 0.9302 | 0.2752 |
| 700–600 | 0.8768 | 0.2530 |
| 600–500 | 0.9021 | 0.2840 |
| 500–400 | 0.9123 | **0.2812** |
| 400–300 | 0.9478 | 0.2815 |
| 300–200 | 0.9568 | 0.2728 |

**Analysis.** As shown in Table C.5, the interval $[\mathbf{900}, \mathbf{800}]$ yields the highest coverage score (**0.9619**).

- **Early Phase ($t \in [900, 800]$):** The diffusion model establishes the global layout and existence of the object. The high coverage with moderate IoU indicates a "semantic blob" that robustly localizes the subject without being rigidly tied to pixel-perfect boundaries. This *plasticity* is crucial for editing, as it allows the model to change the object's pose or shape.
- **Mid-to-Late Phase (e.g., $t \approx 400$):** While IoU peaks around $t = 400$, coverage drops (0.9123). At this stage, the model focuses on fine-grained textures (separating clothes, face, etc.), leading to fragmented maps. High IoU here implies rigid spatial constraints, which would limit the edit to simple texture swapping rather than structural manipulation.

Therefore, we select $[900, 800]$ as the universal probe interval to maximize semantic capture while retaining sufficient flexibility for structural edits.

C.7 COMPARISON WITH DIFFEDIT: MODULATION VS. FILTERING

Although both DiffEdit (Couairon et al., 2023) and Prism-Edit leverage guidance differences to identify semantically meaningful regions, the two methods belong to fundamentally different classes of mechanisms.

**DiffEdit (Latent Filtering).** DiffEdit constructs a binary spatial mask from the guidance magnitude and *overwrites the latent representation* inside the masked region. This latent replacement operation functions as a hard spatial *filter*: regions with weak guidance are entirely removed from the editing process, while strongly activated regions are preserved. Consequently, DiffEdit fails to edit background regions where the guidance signal is naturally weak. This limitation stems from a theoretical

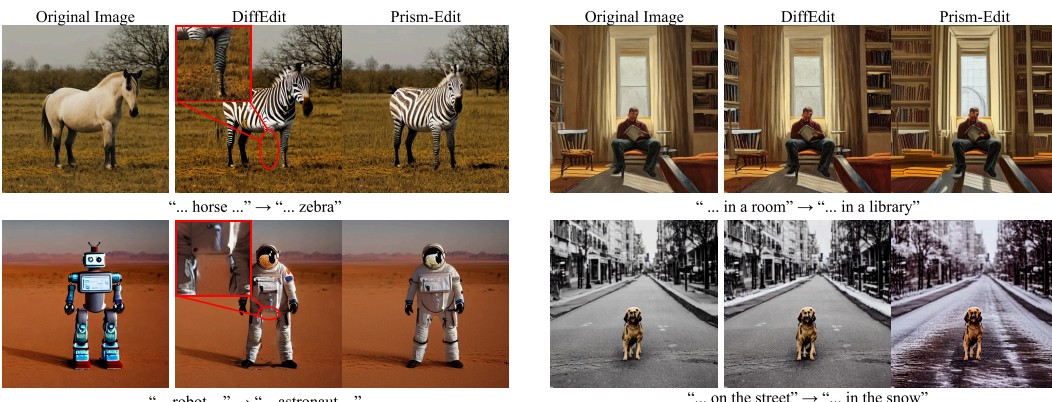

Figure C.10: **Qualitative Comparison with DiffEdit.** We compare Prism-Edit against DiffEdit on both object and background editing tasks using Stable Diffusion v1.5. **(Left Columns - Object Edit):** DiffEdit's hard latent masking introduces severe artifacts and unnatural boundaries (highlighted in red zooms), as it forcibly pastes the edited content. Prism-Edit, using guidance modulation, blends the zebra and astronaut naturally. **(Right Columns - Background Edit):** DiffEdit fails to alter the background ("room" and "street" remain unchanged) because the weak background guidance signals are filtered out by its masking threshold. Prism-Edit successfully amplifies these signals to generate the "library" and "snow" scenes.

oversight: DiffEdit assumes low-magnitude regions are irrelevant. However, as discussed in **Section 4 (Information Imbalance)**, these regions yield weak gradients not because they lack semantic meaning, but because they possess low Fisher information density. By filtering them out, DiffEdit inadvertently discards the valid semantic signals required for background editing.

**Prism-Edit (Guidance Modulation).** In contrast, Prism-Edit never masks or replaces latent variables. It operates exclusively in *guidance space*, applying a semantic weighting to the guidance update $\Delta\epsilon$. Weak but semantically relevant signals are not discarded; instead, they are selectively amplified through Z-score normalization. This effectively counteracts the Information Imbalance, allowing Prism-Edit to forcefully edit low-information regions (backgrounds) that DiffEdit theoretically discards as noise. This preserves structural continuity across the image and enables reliable background editing without introducing the hard spatial artifacts characteristic of latent filtering.

**Quantitative Verification.** We further validated this on a subset of 30 editing tasks using Stable Diffusion v1.5, covering both object and background modifications. Prism-Edit achieved a higher mean CLIP score (**0.2359**) compared to DiffEdit (0.2298), confirming that our modulation approach aligns better with the target prompts while maintaining image naturalness. A visual comparison is provided in Figure C.10.

### C.8 ANALYSIS ON OBJECT-SCARCE SCENES

To validate the universality of the Semantic Scale Hypothesis beyond object-centric images, we extend our analysis to object-scarce domains such as landscapes, fluid textures, and abstract gradients. We find that even in the absence of explicit foreground objects, the guidance magnitude ($||\Delta\epsilon||$) effectively adapts to the **local information density**, separating "implicit structure" from "global atmosphere." As illustrated in Figure C.11, we categorize the behaviors into four distinct patterns:

- **Pattern 1: Localized Color Structures (e.g., Cloudy Sky).** In scenes containing localized structures like cloud patterns or light reflections, these features act as high-information anchors. High-magnitude edits ($\gamma_{high}$) modify these local contrasts, while low-magnitude edits ($\gamma_{low}$) shift the ambient tone or lighting.
- **Pattern 2: Structure vs. Atmosphere (e.g., Ocean).** For scenes mixing smooth masses (e.g., water bodies) with sharp lines (e.g., horizon), the guidance magnitude naturally disentangles them. High-magnitude guidance isolates sharp features for structural edits, whereas low-magnitude guidance alters the global sea state or atmospheric mood.

- **Pattern 3: Implicit Object Hallucination (e.g., Marble).** Strong texture features, such as marble veins, capture high guidance magnitudes. Editing these regions with object-centric prompts often "hallucinates" 3D-like structural changes, treating the veins as pseudo-objects or anchors for new geometry.

- **Pattern 4: Frequency-Based Scaling (e.g., Abstract Gradients).** In edge-dominant images, high-magnitude signals concentrate on high-frequency boundaries, causing sharp transitions at edges. In contrast, low-magnitude signals induce smooth, global color drifts across the flat regions.

**Failure Mode.** In extremely smooth, uniform images (e.g., flat color fields) where high-frequency structure is virtually absent, the information density becomes uniform. In such cases, the semantic separation weakens, resulting in either negligible changes or global monotone shifts. This limitation is consistent with our hypothesis that guidance magnitude relies on information disparity.

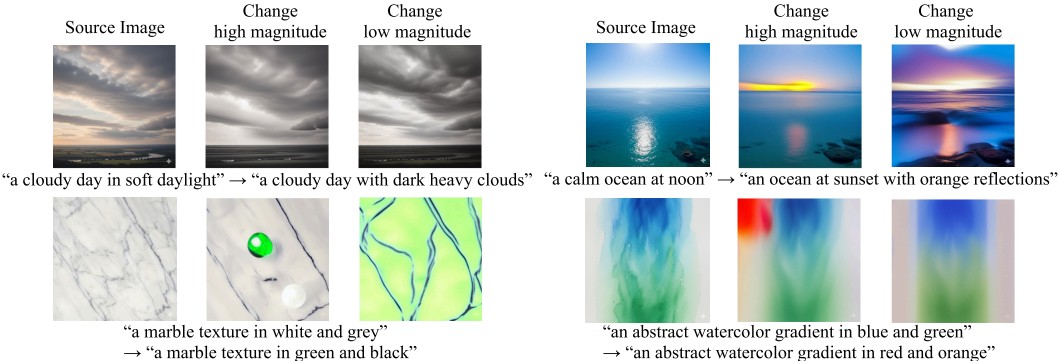

Figure C.11: **Analysis of Semantic Scale in Object-Scarce Scenes.** Even without salient objects, the guidance magnitude separates local structures (e.g., veins, horizons) from the global atmosphere, supporting the universality of our hypothesis.

## C.9 ROBUSTNESS ANALYSIS: INVERSION METHODS AND PROMPT VARIATIONS

To ensure the reliability of our proposed Semantic Scale Hypothesis, it is essential to verify whether the relationship between guidance magnitude and semantic scale holds consistently across different sampling conditions and prompts.

To address this, we conducted a comprehensive robustness analysis by varying two key factors on the same source images:

1. **Inversion Technique:** We compared the standard first-order **DDIM Inversion** with the high-order **DPM-Solver Inversion** (Lu et al., 2022; Hong et al., 2024), utilizing the exact inversion method for the latter.

2. **Target Prompts:** We applied distinct target prompts to the same source image (e.g., editing a "wooden house" into a "stone castle" vs. an "autumn forest") to test how the semantic map responds to different editing intents.

Figure C.12 presents the comparative results.

**Robustness and Adaptivity.** As shown in the histograms and difference maps, we observe two distinct behaviors that validate our hypothesis:

- **Solver Consistency (Nearly Invariant):** Across different inversion solvers (DDIM vs. DPM-Solver), the magnitude distributions exhibit *striking similarity*. Although minor numerical fluctuations exist due to the solver order, the overall distributional shape and the heavy-tail characteristics remain nearly invariant, confirming that the guidance signal is robust to the choice of inversion algorithm.

- **Prompt Adaptivity with Preserved Scaling:** Across different prompts, the distribution *adapts* to the nature of the edit (e.g., structural edits induce a heavier tail than stylistic ones).

Crucially, however, the **spatial scaling principle remains intact**. As visualized in the difference maps, regardless of the prompt, the relative hierarchy is preserved: high-magnitude regions consistently localize the foreground object structure, while the background consistently falls into the low-magnitude range. This demonstrates that while the global energy may shift, the semantic separation logic ($||\Delta\epsilon||_{\text{obj}} \gg ||\Delta\epsilon||_{\text{bg}}$) is never broken.

This confirms that the Semantic Scale Hypothesis is not only robust to solver variations but also correctly reflects the semantic intensity of the text prompt without violating the core information-theoretic scaling law.

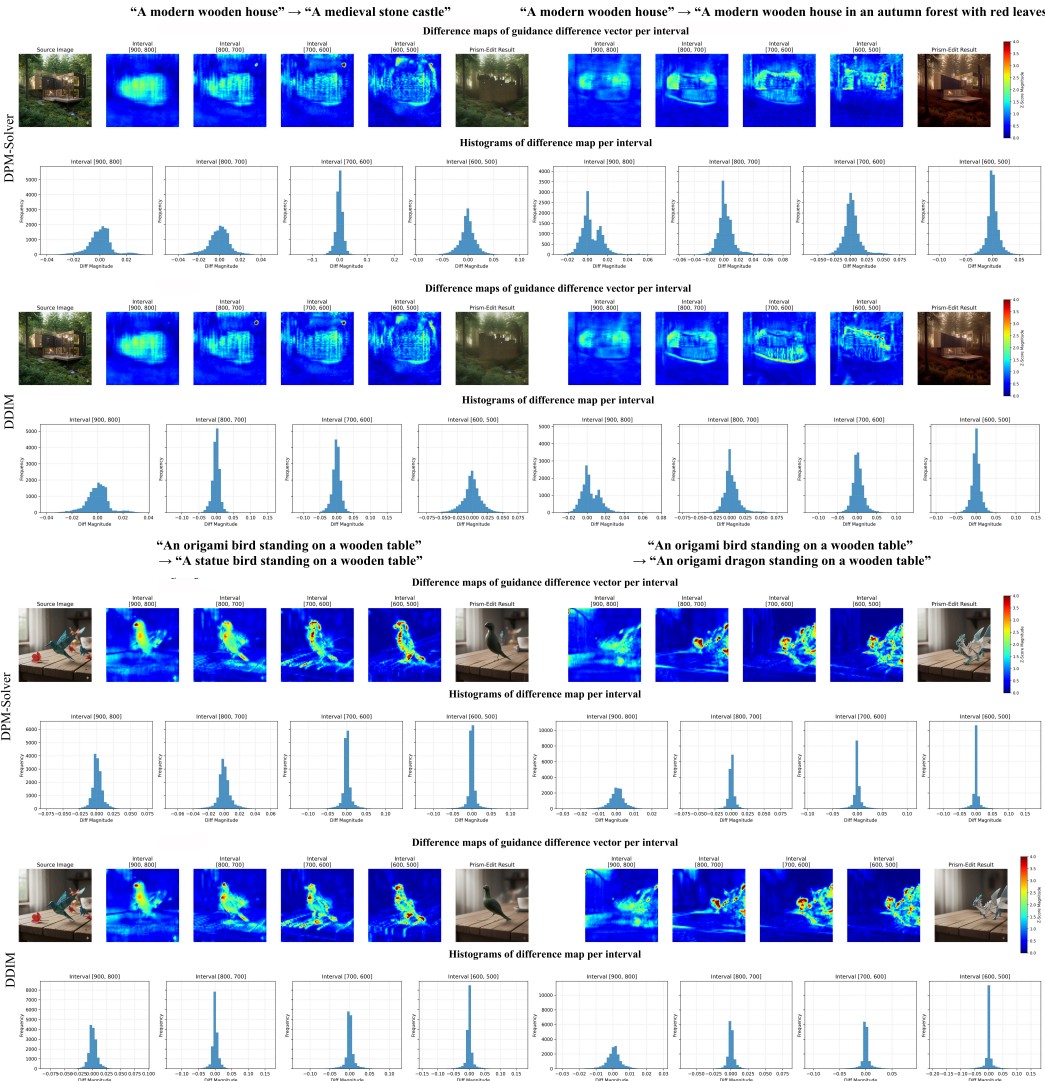

Figure C.12: **Robustness Analysis of the Guidance Difference Vector.** We visualize the histograms and spatial maps of the guidance difference vector magnitudes ($||\Delta\epsilon||$). (1) **Across Inversion Methods**, the distributions are nearly invariant. (2) **Across Target Prompts**, the distributions adapt to the editing task magnitude. However, the spatial maps confirm that the **semantic scaling principle holds**: object regions consistently yield higher magnitudes than backgrounds, regardless of the prompt.

## C.10 Effectiveness of the Static Mask Module

While our main experimental results demonstrate that the *Dynamic Guidance Modulation* alone achieves strong empirical effectiveness in general scenarios, we included the *Static Mask* module

as an optional safety net for corner cases requiring strict preservation. To empirically validate this design choice, we conducted a controlled quantitative and qualitative ablation study.

**Experimental Setup.** We utilized the **CUB-200-2011** dataset (Wah et al., 2011), which contains images with high-frequency background textures (e.g., dense foliage, branches) that are prone to semantic leakage. We randomly selected 100 images from the test set (seed=0) and performed the editing task of transforming birds into "*Wooden Carvings*". This specific prompt was chosen because it involves significant texture and geometric changes that often bleed into the background in standard diffusion editing.

**Quantitative Analysis.** We compared three settings: (1) *Dynamic Only* (Default), (2) *Dynamic + Static Mask* (Optional), and (3) a reference utilizing the Ground Truth (GT) bounding box mask. We measured **CLIP Score** (text alignment) and **SSIM** (background structural similarity).

Table C.6: **Quantitative Ablation on CUB-200.** Comparison of editing modes on 100 samples (Bird → Wooden Carving). The Static Mask mode achieves the highest background preservation (SSIM), surpassing even the GT mask baseline, while the Dynamic mode offers higher editability (CLIP).

| Method | CLIP (Editability) ↑ | SSIM (Preservation) ↑ |
|---|---|---|
| Dynamic Only (Default) | **0.2393** | 0.6792 |
| Dynamic + Static Mask (Optional) | 0.2132 | **0.6896** |
| *(Ref) Dynamic + GT Bbox Mask* | *0.2264* | *0.6859* |

As shown in Table C.6, a clear trade-off exists. The **Dynamic Mode** yields a higher CLIP score, reflecting its flexibility in blending the target concept with the scene. Conversely, the **Static Mask Mode** achieves the highest SSIM, slightly outperforming even the Ground Truth mask baseline. This confirms that the static mask effectively acts as a "deterministic safety net" for pixel-perfect background preservation.

**Qualitative Analysis.** Figure C.13 visualizes the distinct behaviors of the two modes:

- **General Cases:** In scenes with clear separation or simple backgrounds, the Dynamic Mode achieves high-quality editing indistinguishable from the Static Mask mode.

- **Structural Degradation:** In scenes with complex foliage, the Dynamic Mode occasionally struggles to separate the object from the texture, causing the target style (e.g., wood texture) to bleed into the surrounding leaves. The Static Mask fully prevents this leakage.

- **Geometric Hallucination:** We observed that for prompts like "Wooden Carving," the Dynamic Mode sometimes hallucinates contextual objects—for instance, creating a base under a seagull due to the model's prior that carvings typically sit on bases (see Figure C.13, bottom row). The Static Mask successfully suppresses this geometric hallucination, maintaining the original flat terrain.

**Summary.** Consequently, the Static Mask is not merely a redundant component but a strategic tool for handling corner cases. While the Dynamic Mode offers superior flexibility and text alignment for standard scenes, the Static Mask provides a deterministic guarantee for structural preservation. This makes it an essential optional module for users who need to strictly isolate the edit target from challenging, texture-rich backgrounds.

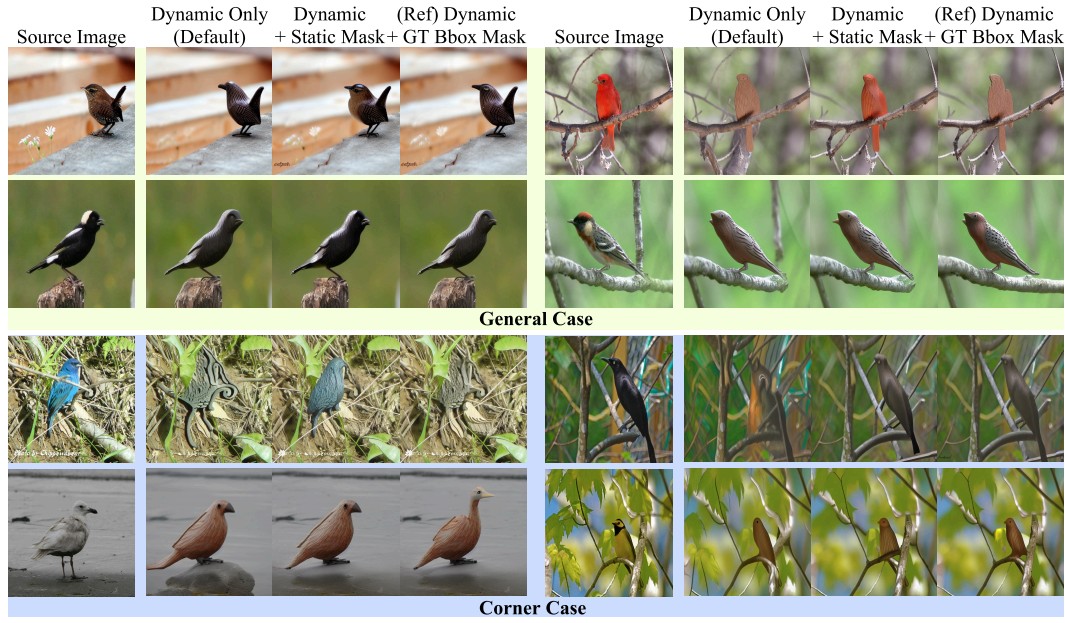

Figure C.13: **Qualitative Ablation on the Static Mask. (Top) General Scenarios:** In scenes with simple backgrounds or clear object separation, the **Dynamic Mode** (Default) achieves high-fidelity editing comparable to the Static Mask mode. **(Bottom) Corner Cases:** In complex scenarios, the Dynamic Mode exhibits specific failure modes: (1) **Structural Degradation:** In the examples with dense foliage (blue, black, and yellow birds), the bird's silhouette is lost as the target texture bleeds into the complex background. (2) **Geometric Hallucination:** In the seagull example, the model hallucinates a new object (a base) under the bird. The **Static Mask** prevents these artifacts by enforcing a strict preservation constraint on non-edit regions.

# D    STATEMENT ON LLM USAGE

In line with the ICLR 2026 policy, we disclose that we used Large Language Models (LLMs) as an auxiliary tool during the preparation of this manuscript. LLM's role was primarily important in enhancing the clarity and readability of the manuscript. It was utilized to refine sentence structure, correct grammatical errors, and improve the overall logical flow of paragraphs, with particular emphasis on the introduction, relevant work, and appendix sections.

