# OpenReview forum: "Deconstructing Guidance: A Semantic Hierarchy for Precise Diffusion Model Editing"
_ICLR.cc/2026/Conference — ICLR 2026 Poster_

### Official Review · Reviewer_6BEq · 2025-10-28

**Soundness:** 2
**Presentation:** 1
**Contribution:** 2
**Rating:** 4
**Confidence:** 4

**Summary:**

This paper introduces semantic scale hypothesis that frames guidance magnitude as an information-theoretic signal to reflect a semantic hierarchy and proposes Prism-Edit that decomposes the guidance signal into different semantic layers for selective control of diffusion model editing.

**Strengths:**

1.	Extensive experiments have been conducted to demonstrate the effectiveness of Prism-Edit, especially for challenging background edits.
2.	The semantic scale hypothesis provides a theoretical perspective to reveal the semantic hierarchy in diffusion model guidance signal.

**Weaknesses:**

1.	Since DiffEdit also leverages guidance differences to achieve selective control in diffusion models, this work’s contribution requires a detailed empirical comparison between DiffEdit’s hard spatial mask and Prism-Edit’s soft binary mask. Without such analysis, the novelty and advancements of the proposed method over previous studies are not sufficiently convincing. Moreover, because Algorithm 1 incorporates a hard-mask module, an ablation study on this component would also be necessary.
2.	A substantial portion of the paper is devoted to deriving closed-form bounds for the guidance difference; however, these results do not seem to be connected to the method’s technical design, nor do they provide deeper insights beyond the analyses already discussed in Sections 4.2 and 4.3.
3.	Some parts of the method are unclear. The choice of probe internal and how to calculate the soft binary mask $W$ lack explanation. Algorithm 2 is unclear.
4.	It miss the introduction of baselines and the details about how to apply Prism-Edit to P2P, PnP and LEDITS++.

**Questions:**

1.	In Fig. 4, Prism-Edit has worse results w.r.t. the CLIP score. Can the authors give some explanation or analysis?
2.	In Fig. 5b, it seems DDIM Inv. obtains better results than Prism-Edits. It would be better to include some analysis for this kind of results.
3.	How does the proposed method utilize negative prompts?
4.	For the different datasets, are the hyperparameter settings for Prism-Edit kept the same?

---

> ### Author Response · Authors · 2025-11-21
>
> We thank the reviewer for the thorough review and constructive criticism. We have carefully addressed your concerns by adding detailed comparisons with DiffEdit, clarifying our method's theoretical grounding, and expanding the experimental analysis.
>
> **W1 — Comparison with DiffEdit & Theoretical Justification**
>
> We fully agree that a comparison is valuable. We have added a detailed visual comparison in **Figure 18 (in Appendix C.7)** and clarified the fundamental theoretical divergence that explains DiffEdit's limitations.
>
> - **DiffEdit's Theoretical Oversight (Binary Masking):** DiffEdit interprets guidance magnitude strictly as a spatial confidence score. It operates on the assumption that low-magnitude regions contain "no editing signal," leading it to filter them out via hard thresholding. **Crucially, it fails to account for the Information Imbalance (Section 4):** background regions inherently yield weak gradients not because they are irrelevant, but because they have **low Fisher information density (high posterior variance)** .
> - **Prism-Edit's Approach (Signal Modulation):** In contrast, Prism-Edit operationalizes the **Semantic Scale Hypothesis**. We recognize that weak signals in the background are valid but information-sparse. Instead of overwriting latents with a mask (which causes artifacts), we **selectively amplify** these weak gradients through Z-score normalization. This allows Prism-Edit to forcefully edit backgrounds that DiffEdit theoretically discards as noise .
> - **Role of Static Mask:** We also clarify that the static mask in Algorithm 1 is merely an optional "safety filter," whereas DiffEdit relies on masking as its primary mechanism.

---

> ### Author Response · Authors · 2025-11-21
>
> **W2 — Connection: Theory to Method**
>
> The theoretical derivation (specifically **Theorem 1** and **Eq. 3**) is not merely descriptive but **prescriptive**—it serves as the mathematical blueprint for our algorithm.
>
> - **Theorem 1 $\to$ Necessity of Z-Score:** Our derivation proves that the raw guidance magnitude $\|\Delta\boldsymbol\epsilon\|$ is strictly bounded by the posterior covariance (inverse Fisher information). This mathematically implies that raw magnitudes are intrinsically biased by local texture complexity (Information Imbalance), making absolute thresholding impossible. **Therefore, Z-score normalization (Eq. 7) is not a heuristic choice but a theoretical necessity** to "cancel out" this variance term and recover the true semantic signal.
> - **Revision:** We have rewritten **Section 5.1** (line 286-) to explicitly map these theoretical bounds to our algorithmic design choices.

---

> ### Author Response · Authors · 2025-11-21
>
> **W3 — Methodological Clarifications (Probe Interval, Mask, Algorithm 2)**
>
> We appreciate the reviewer's detailed scrutiny. We have significantly expanded Section 5 and added Appendix C.6 to remove these ambiguities.
>
> - **Choice of Probe Interval ([900, 800]):** This choice is backed by empirical evidence, not heuristics. As detailed in our new ablation study in **Table 6 (Appendix C.6)**, this specific high-noise interval yields the highest **semantic coverage (Recall: 0.962)** on the COCO dataset. It captures the global structural layout before the model commits to fine-grained textures, making it the optimal window for semantic steering.
> - **Calculation of Soft Binary Mask ($W_{sem,t}$):** We clarified in **Section 5.2** that $W_{sem,t}$ is derived directly from the Z-normalized guidance magnitude. In practice, we apply the specific thresholds (e.g., $>3.0\sigma$ for objects) defined in **Table 1** to binarize this map. This ensures stable region selection compared to continuous weighting.
> - **Algorithm 2 (Refinement):** We have rewritten the description of Algorithm 2 to be explicit. It implements a standard **morphological closing** operation (dilation followed by erosion) to ensure that the extracted semantic regions are spatially contiguous and free of high-frequency noise holes.

---

> ### Author Response · Authors · 2025-11-21
>
> **W4 — Introduction of Baselines (P2P, PnP, LEDITS++)**
>
> Thank you for pointing this out. We have added a new section, **Appendix B.3**, which details how Prism-Edit is integrated with each baseline.
>
> - **Mechanism:** Since Prism-Edit operates on the guidance vector, it serves as a universal plug-and-play module.
> - **Integration:** For P2P/PnP, we replace the standard CFG term. For LEDITS++, we specifically exclude its native quantile-masking step to demonstrate the superiority of our gradient modulation.

---

> ### Author Response · Authors · 2025-11-21
>
> **Q1 & Q2 — CLIP Score & DDIM Inversion**
>
> We thank the reviewer for these insightful observations. We agree that CLIP scores require careful interpretation.
>
> - **The Trade-off:** CLIP metrics are biased towards global modifications. Baselines like DDIM Inversion often alter the entire scene (global drift) to match the prompt, inflating CLIP scores but degrading identity.
> - **Evidence:** This trade-off is illustrated in **Figure 5**. While CLIP scores may plateau, **Figure 5(a)** demonstrates that Prism-Edit achieves a significantly higher **DINO/SSIM ratio**, confirming that our method prioritizes **local semantic fidelity** over global drift.
> - **Revision:** Although we briefly touched upon this phenomenon in our original submission, your feedback highlighted the need for a more rigorous explanation. So, we have updated **Section 6.1** to explicitly discuss this trade-off.

---

> ### Author Response · Authors · 2025-11-21
>
> **Q3 — Negative Prompts**
>
> We clarify our negative prompt strategy:
>
> - **Inversion Phase:** We use an **empty string ("")** to ensure faithful reconstruction.
> - **Editing Phase:** We use the **source prompt ($p_{\text{src}}$)** as the negative prompt to effectively neutralize the original semantic concept while guiding towards the target.
> - **Revision:** These details are now included in **Appendix C.4**.

---

> ### Author Response · Authors · 2025-11-21
>
> **Q4 — Hyperparameter Consistency**
>
>
> We use the same settings (probe window [900–800], z-score thresholds) across all datasets. This robustness is enabled by the scale-invariance of our Z-score normalization.

---

> > ### Comment · Reviewer_6BEq · 2025-11-25
> >
> > Thanks for the detailed response. However, some concerns are remained as below:
> >
> > 1. Weakness 1 is not fully addressed. “Algorithm 1 incorporates a hard-mask module, an ablation study on this component would also be necessary”
> > 2. Although the authors claim that Theorem 1 and Eq. (3) underpin Eq. (7), this technique constitutes only a minor part of the paper. Moreover, Eq. (7) is fairly intuitive—normalization is naturally required when setting thresholds across different distributions. I would instead expect the extensive theoretical analysis to provide a foundation for the semantic-scale hypothesis or for Section 5.2, the main technical contribution of the paper.
> > 3. Inconsistent setting: Algorithm 1 of the original submission shows using probe interval [900, 880], but now the ablation study indicates [900,800].
> > 4. Applying Prism-Edit to P2P is mentions only at the final editing sampling step, not consistent with Algorithm 1.

---

> > > ### Author Response · Authors · 2025-11-26
> > >
> > > **3. Inconsistency in Probe Interval ([900, 880] vs [900, 800])**
> > >
> > > You are correct that the original submission used [900, 880]. We clarify that this is not an accidental inconsistency but a **transparent update** driven by our new experimental findings.
> > >
> > > * **Transparency:** As explicitly noted in **Item 4 of the "Summary of Revisions"**, we refined this interval based on the rigorous ablation study requested by reviewers (Table 6 in Appendix C.10).
> > > * **Refinement:** We found that extending the window to **[900, 800]** achieves the highest semantic coverage (Recall: 0.962) while maintaining structural plasticity, making it the optimal "sweet spot" compared to the narrower original setting.
> > > * **Action Taken:** We have updated Algorithm 1 and Section 5.1 to reflect this optimized setting. Importantly, this refinement enhances robustness but **does not alter the fundamental qualitative or quantitative conclusions** reported in our initial submission.
> > >
> > > **4. Integration with P2P (Algorithm consistency)**
> > >
> > > **We thank you for pointing out the ambiguity regarding baseline integration.** We acknowledge that this confusion arose because **Algorithm 1 describes the standalone Prism-Edit pipeline and is not meant to represent the hybrid P2P variant.**
> > >
> > > * **Clarification:** When integrating with baselines like P2P, Prism-Edit functions as a plug-and-play module. We respect P2P's original cross-attention control to preserve layout and intervene *only* at the **final guidance computation step** of each sampling iteration, overriding the standard CFG formula:
> > >     $$
> > >     \epsilon_{\text{final}} = \epsilon_{\text{uncond}} + \gamma \cdot (\epsilon_{\text{P2Pcond}} - \epsilon_{\text{uncond}}) \odot W_{\text{sem}}
> > >     $$
> > > * **Action Taken:** To resolve this confusion, we have added a dedicated section, **Appendix B.3 (Baseline Integration Details)**, in the revised manuscript. This section explicitly specifies that we employ **Dynamic Guidance Modulation** only (without the static mask) for baseline integration, clearly distinguishing the hybrid workflow from the standalone algorithm.
> > >
> > > We hope these clarifications address your concerns, and we sincerely appreciate your constructive engagement. We have reflected all these points, including the new ablation study in **Appendix C.10** and **Figure 21**, in the revised version.

---

> ### Author Response · Authors · 2025-11-26
>
> We are truly grateful for your dedicated and enthusiastic engagement with our work. We consider ourselves fortunate to have received such rigorous feedback, which has played a pivotal role in refining both the theoretical clarity and technical consistency of our manuscript.
>
> We address your remaining concerns point-by-point below.
>
> **1. On the Necessity of an Ablation Study for the "Hard-Mask Module"**
>
> We appreciate your suggestion. While our main results (Tables 3-5, Figures 6-7) were generated using the **"Dynamic Guidance Modulation"** mode solely (proving its standalone robustness), we agree that evaluating the specific contribution of the static mask clarifies the method's design.
>
> To address this, we conducted a **controlled quantitative and qualitative ablation study** on the **CUB-200-2011** dataset. We randomly selected **100 images** from the test set (seed=0) and performed the editing task of transforming birds into **"Wooden Carvings"**, a prompt known to cause significant texture leakage into complex backgrounds.
>
> **(1) Quantitative Evaluation:**
> We compared our **Dynamic Mode**, **Static Mask Mode**, and a **Ground Truth (GT) Mask** reference.
>
> | Mode | CLIP (Editability) $\uparrow$ | SSIM (Preservation) $\uparrow$ |
> | :--- | :--- | :--- |
> | **Dynamic Only (Default)** | **0.2393** | 0.6792 |
> | **Dynamic + Static Mask** | 0.2132 | **0.6896** |
> | *(Ref) Dynamic + GT Bbox Mask* | *0.2264* | *0.6859* |
>
> * **Preservation:** The **Static Mask mode achieves the highest SSIM**, outperforming the Dynamic mode and even slightly surpassing the Ground Truth mask baseline. This confirms its efficacy as a "safety net" for pixel-perfect preservation.
> * **Editability:** The Dynamic mode yields a higher CLIP score, reflecting its flexibility in blending the target concept. **However, this unconstrained flexibility acts as a double-edged sword: while it allows better concept integration, it also poses a risk of semantic leakage (lower SSIM) in complex backgrounds, which the Static Mask effectively mitigates.**
>
> **(2) Qualitative Analysis (Figure 21 in Revised PDF):**
> As visualized in the newly added **Figure 21**, we identified critical failure modes in the Dynamic Only setting that the Static Mask successfully resolves:
> * **General Case:** In standard scenes, both modes produce identical high-quality edits.
> * **Structural Degradation (Corner Case 1):** In scenes with dense foliage (e.g., the blue and yellow bird examples), the Dynamic Mode struggles to separate the object from the background, causing the **bird's shape to dissolve** or bleed into the complex texture. The Static Mask effectively preserves the subject's boundary.
> * **Geometric Hallucination (Corner Case 2):** Notably, for prompts like "Wooden Carving," the Dynamic Mode sometimes **hallucinates contextual objects** (e.g., creating a base under the seagull due to the prior that "carvings have bases"). The **Static Mask** successfully suppresses this geometric hallucination, maintaining the original flat terrain.
>
> **Conclusion:** This trade-off justifies our design: Dynamic Modulation serves as the balanced default, while the Static Mask is essential for tasks requiring strict structural guarantees. We wrote this module as **Optional** in Algorithm 1.

---

> ### Author Response · Authors · 2025-11-27
>
> **2. Connection between Theory (Theorem 1) and Method (Sec 5.2)**
>
> We respectfully argue that Theorem 1 is not merely a post-hoc motivation, but the **mathematical constraint that directly prescribes the structure of Algorithm 1.** The connection proceeds in two logical steps:
>
> **(A) Theorem 1 $\rightarrow$ Necessity of Z-Score Normalization (Decoupling Semantics from Variance)**
> Theorem 1 establishes that the raw guidance magnitude $||\Delta\epsilon||$ is not a pure semantic quantity but is intrinsically coupled with the inverse posterior variance $1/\sigma_t$ (local information density).
> * **The Problem (Information Imbalance):** This coupling creates a systematic bias where high-variance regions (backgrounds) yield vanishingly small magnitudes, while low-variance regions (object cores) yield disproportionately large ones. Consequently, raw magnitudes cannot be thresholded consistently across different image contexts.
> * **The Algorithmic Solution:** To recover the latent semantic signal, we must **decouple** this variance term. **Z-score normalization (Eq. 7)** is the precise operation required to remove this variance-induced bias:
>     $$
>     z = \frac{||\Delta\boldsymbol\epsilon|| - \mu}{\sigma}
>     $$
>     This transforms the raw gradients into a **scale-invariant semantic signal**, which is why Algorithm 1 *must* begin with normalization to establish a stable semantic hierarchy.
>
> **(B) Theorem 1 $\rightarrow$ Necessity of Modulation (Signal Restoration)**
> Theorem 1 further reveals that weak gradients in background regions are **not noise**, but **valid semantic signals suppressed by high variance**.
> * **The Problem:** Previous methods like DiffEdit incorrectly interpret these small magnitudes as "irrelevant" and filter them out.
> * **The Algorithmic Solution:** Our theory dictates the opposite approach: if the signal attenuation is caused by high variance, the correct counter-operation is **variance-compensating amplification**. This directly motivates our **Dynamic Modulation** mechanism in Section 5.2:
>     $$
>     \Delta\boldsymbol\epsilon_{\text{mod}} = \gamma \cdot W_{\text{sem}} \odot \Delta\boldsymbol\epsilon
>     $$
>     Here, amplification is not a heuristic tweak but an **algorithmic inversion** of the suppression predicted by Theorem 1, restoring the editability of low-information regions. **This theoretical insight legitimizes our approach of multiplying these weak signals by large factors ($\gamma \approx 30\sim 40$) to recover the intended semantics, a step that would be heuristically risky without this grounding.**
>
> **Summary of Theoretical-Algorithmic Mapping:**
>
> | **Theorem 1 Insight** | **Algorithmic Design (Sec 5.2)** |
> | :--- | :--- |
> | Magnitude is biased by local variance ($1/\sigma_t$) | Apply **Z-score normalization** to decouple variance |
> | Weak background signals are valid but suppressed | Apply **Amplification ($\gamma$)** to restore signal strength |
> | Semantic layers are obscured in raw space | **Weight map ($W_{sem}$)** is constructed in Z-space |

---

### Official Review · Reviewer_sjg1 · 2025-10-30

**Soundness:** 3
**Presentation:** 3
**Contribution:** 3
**Rating:** 6
**Confidence:** 3

**Summary:**

This paper suggests the Semantic Scale Hypothesis: the magnitude of the guidance vector correlates with the semantic scale of an edit. Specifically, low-variance regions, where the model is more certain and which typically correspond to foreground objects, exhibit higher guidance magnitudes, while high-variance regions, often background, show lower magnitudes. Building on this, the authors propose Prism-Edit, which adjusts edits by selecting ranges of guidance magnitudes, enabling object-only or background-only manipulation without requiring any object masks. The method is shown to work across various types of models and editing techniques.

**Strengths:**

- Clear conceptual novelty. To my knowledge, this is the first work to analyze editing behavior through the magnitude of the guidance vector itself. Furthermore, the analysis that higher-magnitude regions align with more certain, typically object-centric areas, while lower magnitudes map to uncertain, often background regions, is both intuitive and seems to have its own advantages

- Mask-free, plug-and-play editing with broad model coverage. By selecting guidance-magnitude ranges, Prism-Edit enables foreground/background-targeted edits without segmentation masks. The procedure is training-free and plug-and-play, making it easy to drop into existing pipelines. Moreover, successful applications across models and editing methods indicate a model-invariant control axis, improving reliability and external validity.

**Weaknesses:**

- Condition sensitivity. Guidance vectors change with prompts, seeds, schedulers, and editing method choices. It is unclear whether the proposed magnitude, semantics relationship consistently holds under such variations. Is this tendency of the relation between guidance magnitude and the semantic scale of edit preserved regardless of the given condition?  Additional analysis would strengthen the claim.

- Thresholding and disentanglement. Thresholds are manually tuned and depend on the editing method (Table 2, Appendix). Figure 10 also suggests continuous trade-offs as the threshold varies, implying object and background information remain partially entangled. While the paper achieves useful partial disentanglement, the dependence on thresholds should be better characterized.

- Per-image statistics & efficiency. The method appears to compute per-image guidance-magnitude statistics. Even with z-score normalization, distributions may differ across images, raising concerns about the speed and stability of edited results depending on the given image and condition.

**Questions:**

- Object-scarce scenes. The paper frames “semantic scale” mainly as object vs. background. How does the interpretation extend to images without salient objects (e.g., textures, landscapes, or abstract scenes)? What do high- vs. low-magnitude regions represent in such cases?

- Magnitude variability. How much do guidance-magnitude distributions vary across images, prompts, and inversion methods? Including representative histograms for multiple settings would clarify variance and help practitioners choose thresholds.

---

> ### Author Response · Authors · 2025-11-21
>
> We greatly appreciate the reviewer's recognition of our work's conceptual novelty and practical value. Your insightful questions regarding condition sensitivity and object-scarce scenes have significantly strengthened our analysis.
>
>
> **W1 & Q2 — Stability Across Inversion Methods, Prompts, and Images**
>
> To demonstrate the robustness of the Semantic Scale Hypothesis, we conducted a comprehensive analysis varying inversion methods, target prompts, and source images. Figure 20 (in Appendix C.9) presents the results:
>
> - **Robustness to Inversion Methods (DDIM vs. DPM-Solver):** We compared DDIM (1st-order) and DPM-Solver (high-order). Despite their algorithmic differences, the semantic maps and magnitude histograms at the early stage ($t \in [900, 800]$) remain **nearly invariant**, capturing identical structural regions. This confirms that Prism-Edit relies on an intrinsic property of the diffusion process (Information Density) rather than artifacts of a specific sampler.
> - **Consistency Across Prompts:** As shown in the bottom rows of **Figure 20**, the semantic map for the same source image remains spatially consistent regardless of the target prompt (e.g., "Castle" vs. "Autumn Forest"). While the histogram distribution adapts to the edit's complexity, the **relative hierarchy** (Object $>$ Background) is strictly preserved.
> - **Conclusion:** Our Z-score normalization effectively aligns these varying raw distributions into a unified control space, ensuring stability across diverse conditions.

---

> ### Author Response · Authors · 2025-11-21
>
> **Q1 — Generalization to Object-Scarce Scenes**
>
> This is an insightful question. As the reviewer noted, "Semantic Scale" extends beyond objects. We analyzed various object-scarce scenarios (e.g., Cloud, Ocean, Marble) in Figure 19 (in Appendix C.8) and identified distinct behaviors:
>
> - **High Magnitude:** Maps to **Local Structures** (e.g., horizon lines, wave crests, marble veins). Edits here act as "structural anchors" or pseudo-objects.
> - **Low Magnitude:** Maps to **Global Atmosphere** (e.g., lighting, overall color tone). Edits here change the "vibe" without destroying geometry.
>
> **Failure Mode (Limitation):**
> We also identified a boundary condition. In extremely smooth, uniform images (e.g., flat color fields or featureless fog) where high-frequency structure is virtually absent, the information density becomes uniform. In such cases, the semantic separation weakens, resulting in either negligible changes or global monotone shifts. This limitation is consistent with our hypothesis that guidance magnitude relies on **information disparity**.
>
> **Conclusion:** Even without explicit objects, the guidance magnitude effectively disentangles local variance from global consistency, provided that sufficient structural variance exists in the source image.

---

> ### Author Response · Authors · 2025-11-21
>
> **W2 & W3 — Efficiency and Overhead**
>
> - **Computation:** Calculating the mean ($\mu$) and standard deviation ($\sigma$) of the guidance tensor takes **$< 1$ms** on a standard GPU.
> - **Impact:** It introduces virtually **zero overhead** to the inference pipeline while ensuring critical stability against image contrast variations.

---

> ### Comment · Area_Chair_vWfb · 2025-11-27
> **Discussion**
>
> Hello, Reviewer sjg1! Thank you for contributing a helpful review. It would be good if you could write a response to the reviewers to discuss the rebuttal. For example, did the rebuttal address your concerns, are there additional questions, … It would be good if you could post something very soon, so that the authors have a chance to respond.

---

### Official Review · Reviewer_jMi2 · 2025-10-31

**Soundness:** 3
**Presentation:** 3
**Contribution:** 2
**Rating:** 4
**Confidence:** 3

**Summary:**

This paper studies the guidance mechanism of the diffusion models, and proposes a hypothesis that the magnitude of the guidance difference vector directly encoded the semantic scale of the edits. Low-variance regions yield large magnitude differences with structural changes, while high-variance regions yield small magnitude differences with stylistic adjustments. By using this observation, it proposes to apply either a dynamic mask or a static mask computed from the guidance magnitude to the original guidance difference, then to perform edit. The results show that the proposed method can do the edit while preserving the layout better than the baselines.

**Strengths:**

1. It’s interesting to mathematically show that the magnitude of the guidance difference itself contains useful information that can be useful to guide the edits.

2. The proposed method can be applied to different editing methods in a plug-and-play way, which is flexible and universal.

**Weaknesses:**

While after applying the proposed method the results can be improved over the original editing method, the edited results are still not very satisfactory. I understand that the results are also greatly influenced by the base model / base method. Just to say in Figure 7, where the base model is advanced, the results after applying the proposed method are not good. With SD3, especially for the bear example, the background looks unrealistic. With Flux RF-Inversion, the background improves, but the overall style goes from realistic to more animation style. With Flux Stable-flow, the foreground objects either have a boundary shadow, or deteriorate.

While I do agree that the proposed method is simple and can improve over the baselines, it seems that the proposed method itself is still not able to achieve very high quality results without any other modifications. This makes me doubt that either there is a better way to utilize the hypothesis rather than simply applying a static/soft mask, or the proposed hypothesis itself alone is not able to achieve a better quality.

**Questions:**

1. What is $M_{final}$ in Equation 8?

2. What is $W_{sem,t}$ in Equation 9? Is it the absolute value of the guidance difference?

3. Is the high-noise window in line 275 needed to be calculated per editing example, or is there a universal $t$ value that can work across every example?

---

> ### Author Response · Authors · 2025-11-21
>
> We sincerely thank the reviewer for the detailed feedback and for acknowledging the mathematical interest and flexibility of our method. We address your concerns regarding visual quality and provide specific clarifications below.
>
> **W — Visual Quality Concerns (Fig. 7)**
> We acknowledge that advanced backbones like SD3 or Flux may introduce their own stylistic artifacts. However, we emphasize:
>
> * **Role:** Prism-Edit is a **semantic steering mechanism**, not a generator. We control *where* the edit occurs; the fidelity comes from the backbone.
> * **Cause of Artifacts (Sensitivity to Prompt Omission):** We observed that these visual shifts often stem from the backbone faithfully reflecting the **absence of specific descriptors**, relying instead on its internal interpretation.
>     1.  **Content (Figure 19):** As seen in the Ocean scene, simply removing the word **"calm"** caused the background waves to become rougher.
>     2.  **Style (Figure 7):** Analogously, without explicit keywords like **"photorealistic,"** the model analyzes the image and generates the object in the style it **considers closest** to the visual context, rather than strictly adhering to realism.

---

> ### Author Response · Authors · 2025-11-21
>
> **Q1 — Definition of $M_{final}$ (Eq. 8)**
>
> $M_{final}$ is the binary static mask used specifically for the optional "Static Mask Blending" mode (Algorithm 1, Lines 7-15). Unlike DiffEdit's hard filtering, this mask acts merely as a permissive safety filter to prevent edits from drifting into irrelevant regions.
>
> **Q2 — Definition of $W_{sem,t}$ (Eq. 9)**
>
> $W_{sem,t}$ is derived from the z-scored $|\Delta\boldsymbol\epsilon_t|$. In practice, we binarize this map to ensure stability and alignment with the Semantic Scale Hypothesis.
>
> - **Clarification:** Although theoretically continuous, in practice, we **binarize** this map (using the thresholds $\ge 3.0\sigma$ for objects, $< 0.6\sigma$ for backgrounds) to ensure stability and strictly align with the Semantic Scale Hypothesis. We have explicitly updated **Section 5.2** to reflect this binarization step.

---

> ### Author Response · Authors · 2025-11-21
>
> **Q3 — Universality of High-Noise Window**
>
> Yes, the window is universal.
>
> - **Evidence:** We performed a new ablation study on the COCO dataset, reported in **Table 6 (Appendix C.6)**.
> - **Result:** The **[900, 800]** interval consistently achieves the highest **Coverage (0.962)** across diverse images. This proves that the global semantic layout is reliably established in this early phase for all samples, confirming that per-image tuning is unnecessary.

---

> > ### Comment · Reviewer_jMi2 · 2025-11-28
> >
> > Thanks for the authors for the details responses to my concerns and other reviewers' concerns. The newly added experiments are very helpful. I don't have other questions so far, and I will update my score.

---

> ### Comment · Area_Chair_vWfb · 2025-11-27
> **Discussion**
>
> Hello, Reviewer jMi2! Thank you for contributing a helpful review. It would be good if you could write a response to the reviewers to discuss the rebuttal. For example, did the rebuttal address your concerns, are there additional questions, … It would be good if you could post something very soon, so that the authors have a chance to respond.

---

### Official Review · Reviewer_5VkF · 2025-11-01

**Soundness:** 3
**Presentation:** 4
**Contribution:** 3
**Rating:** 4
**Confidence:** 4

**Summary:**

The paper proposes the Semantic Scale Hypothesis which motivates a training-free, plug-and-play module to improve image editing performances across models.

The paper could be a nice contribution on understanding the diffusion process. I'm willing to increase my score if the questions are clarified.

**Strengths:**

- The paper is well-written
- Strong empirical support and a nice theoretical connection for the Semantic Scale Hypothesis
- The method is model-agnostic

**Weaknesses:**

- The thresholds used in Table 1 seem different for each baseline method, but "fixed thresholds" are claimed to be stable (286-288). The guidance scales seem to be different as well. How sensitive are these hyperparameters? How should a user pick the good ones?

**Questions:**

- Why is the noise range $t \in [900,880]$. Does a fixed $t=900$ or a wider range $t \in [900,800]$ not work? How to justify the number $880$?
- What is the motivation behind $W_{sem,t}$ as it's computed from $\Delta_{\epsilon_t}$ itself?

---

> ### Author Response · Authors · 2025-11-21
>
> We sincerely thank the reviewer for the constructive feedback and for recognizing the strong empirical support and theoretical connection of our work. We address your questions regarding hyperparameters and theoretical motivations below.
>
> **W1 — Threshold Sensitivity & Stability**
>
> We clarify a crucial distinction: **Thresholds are baseline-dependent, but edit-independent.**
>
> - **Reason:** Each backbone (e.g., SD1.5 vs. Flux) has a distinct information variance schedule, leading to different raw $|\Delta\boldsymbol\epsilon|$ distributions.
> - **Stability:** However, **within a given baseline**, a single set of thresholds works robustly across diverse prompts, seeds, and editing tasks. This validates our Semantic Scale Hypothesis: the *relative* semantic ordering (background $\to$ mid-level $\to$ object core) remains invariant, even if the absolute scale shifts between architectures. Thus, users do not need to tune thresholds for individual images.

---

> ### Author Response · Authors · 2025-11-21
>
> **Q1 — Why the probe window [900–880]?**
>
> Thank you for raising this important point. To justify this choice, we performed a rigorous quantitative ablation on the COCO "person" class (highly deformable objects). The results are reported in **Table 6 (in Appendix C.6)**:
>
> - **[900–880] (Too Narrow):** While this early window captures structure, the interval is too short to accumulate a stable signal, yielding a lower Coverage (0.9109).
> - **[900–800] (Ours, Optimal):** Extending the window to $t=800$ achieves the **highest Coverage (0.962)** with moderate IoU (0.265). This interval is sufficient to robustly capture the "semantic whole" of the object while maintaining structural plasticity.
> - **[500–400] (Too Rigid):** While IoU increases (0.281), the coverage drops and the guidance becomes spatially rigid. This forces the model to only perform texture swaps rather than structural edits.
>
> **Conclusion:** **Prompted by your insightful suggestion**, we verified that **[900–800]** is the universal "sweet spot" where the model defines the global semantic structure before committing to fine-grained details.

---

> ### Author Response · Authors · 2025-11-21
>
> **Q2 — Justification for Z-Score Normalization**
>
> The motivation is grounded in the Information Imbalance principle (Section 4).
>
> - **The Problem:** As evidenced by the histograms in **Figure 20 (in Appendix C.9)**, while the *distributional shape* (heavy tail for structure) is consistent across edits, the **exact numerical range** and statistical properties ($\mu, \sigma$) of $|\Delta\boldsymbol\epsilon|$ vary depending on the image content and the specific backbone model. Consequently, a fixed absolute threshold (e.g., $>0.1$) is brittle and cannot generalize.
> - **The Solution:** Z-score normalization aligns these distributions into a unified control space. By rescaling gradients based on their instance-specific statistics, we convert raw values into a **relative significance score**. This ensures that a threshold of $>3\sigma$ always robustly captures the high-information "object core," regardless of the slight variations in raw signal strength.

---

> > ### Comment · Reviewer_5VkF · 2025-11-26
> >
> > Thank you for the detailed responses and the additional ablation study. I have updated my score.

---

### Author Response · Authors · 2025-11-23
**Summary of Revisions and Additional Experiments**

We sincerely thank all reviewers for their insightful and constructive feedback. We are encouraged by the positive reception regarding the novelty of the **Semantic Scale Hypothesis** and the model-agnostic nature of **Prism-Edit**.

In response to your valuable suggestions, we have significantly revised the manuscript. The **major** updates are summarized below:

**1. Robustness Analysis: Inversion Methods & Guidance Histograms (Appendix C.9, Figure 20)**
* To address concerns regarding condition sensitivity (**Reviewer sjg1**) and the rationale for Z-score normalization (**Reviewer 5VkF**), we added **Appendix C.9**.
* We visualized the histograms of the guidance difference vector magnitudes ($\|\Delta\boldsymbol\epsilon\|$) across different inversion methods (DDIM vs. DPM-Solver) and prompt variations.
* **Result:** The distributional shape remains similar across methods, but the variance differs substantially, thereby validating our normalization strategy.

**2. Generalization to Object-Scarce Scenes (Appendix C.8, Figure 19)**
* Following **Reviewer sjg1**'s suggestion, we analyzed the method's behavior on texture and landscape images lacking salient objects.
* **Result:** Prism-Edit effectively disentangles **"Local Structure"** (high magnitude) from **"Global Atmosphere"** (low magnitude), demonstrating utility beyond object-centric editing.

**3. Theoretical Comparison with DiffEdit (Appendix C.7, Figure 18)**
* To clarify the advancement over DiffEdit (**Reviewer 6BEq**), we added a detailed comparison in **Appendix C.7**.
* We theoretically frame DiffEdit as a **"Latent Filtering"** approach that fails due to information imbalance (filtering out low-density regions), whereas Prism-Edit is a **"Signal Modulation"** approach that selectively amplifies weak but valid background signals.

**4. Quantitative Ablation on Probe Interval (Appendix C.6, Table 6)**
* To justify the probe interval selection (**Reviewer 5VkF**), we performed a rigorous ablation study on the COCO dataset.
* **Result:** We identified that extending the window from $[900,880]$ to **$[900, 800]$** achieves the highest semantic coverage (**Recall: 0.962**) while maintaining structural plasticity, establishing it as the universal "sweet spot" compared to narrower intervals.

**5. Baseline Integration Details (Appendix B.3)**
* We added **Appendix B.3** to provide specific details on how Prism-Edit integrates with **P2P, PnP, and LEDITS++** (**Reviewer 6BEq**).
* We clarify that for LEDITS++, we explicitly **exclude** its native masking step to demonstrate the effectiveness of our gradient modulation.

**6. Methodological Clarifications (Section 5, Appendix B.1)**
* To address requests for greater clarity regarding our algorithm (**Reviewer 6BEq**, **Reviewer jMi2**), we have refined **Section 5**.
* We explicitly defined the binarization step for the dynamic weight map $W_{sem,t}$ and clarified the role of the static mask as an optional safety filter.
* **Result:** The updated **Algorithm 1** and **Algorithm 2** now provide a precise, reproducible description of the modulation process.

**7. Clarification on Evaluation Metrics & Trade-offs (Section 6.1)**
* To address concerns regarding **quantitative metrics (CLIP score)** (**Reviewer 6BEq**), we explicitly discussed the inherent trade-off between **disentanglement** (local fidelity) and **global alignment** (global modification).
* We clarify that while baselines may achieve higher CLIP scores by altering the entire scene (global drift), **Prism-Edit prioritizes the strict preservation of unedited regions**, as evidenced by superior **DINO/SSIM** ratios. We also note that certain visual shifts (**Reviewer jMi2**) often stem from the backbone's **sensitivity to prompt omissions** (e.g., absence of "photorealistic"), which causes the model to fall back on its internal stylistic priors. Such shifts become noticeable specifically because Prism-Edit preserves the **original context of the images**, whereas baselines typically reshape the global style in a way that masks these inconsistencies.


We believe these additional experiments and clarifications strongly reinforce the validity and robustness of our proposed method. We look forward to further discussion.

---

### Author Response · Authors · 2025-11-30
**Summary of Revisions and Reviewer Consensus (To the New Area Chair)**

**Dear Area Chair,**

We sincerely understand the unusual situation caused by the recent platform issues.
To help with your assessment, we have summarized the progress made during the rebuttal and discussion period, **before the score reversion**.

---

## 1. Reviewers Who Explicitly Indicated Score Increases

Before the discussion period was halted, **two reviewers clearly stated that they raised / would raise their scores**, based on the rebuttal and the additional materials we provided.

### • Reviewer 5VkF (Originally 4 → 6)
> *"I have updated my score."* (Nov 26)

The reviewer noted that their concerns about Z-score normalization and the interval of Static Masks were fully addressed.

### • Reviewer jMi2 (Originally 4 → at least 6)
> *“…responses to my concerns **and other reviewers’ concerns**… I will update my score.”* (Nov 28)

This reviewer revised their opinion after reading our answers not only to their own questions, but also to those raised by other reviewers suggesting confidence in the overall consistency of our clarifications.

---

## 2. Additions and Revisions Made During Rebuttal

Several important analyses were added in direct response to reviewer comments, particularly from **Reviewer 6BEq** and **Reviewer sjg1**, who asked for more detailed evidence.

### • DiffEdit comparison & Static Mask ablation (Appendix C.7, C.10)
In response to **Reviewer 6BEq**, we added:

- a detailed comparison with DiffEdit,
- a new ablation of the Static Mask module,
- and an additional reference using ground-truth bounding boxes.

These results clarify our Dynamic Guidance Modulation and the role of the Static Mask as a safeguard in challenging scenes.

### • Object-scarce scenes (Appendix C.8)
Addressing **Reviewer sjg1’s** question, we analyzed images without clear foreground objects.
High-magnitude regions correspond to structural cues (e.g., horizon lines), while low-magnitude regions reflect global appearance.
We also pointed out the boundary cases where structure is nearly absent.

### • Theoretical clarification (Theorem 1 → Algorithm 1)
To address concerns from **Reviewer 6BEq**, we made the connection between Theorem 1 and our algorithmic steps more explicit, explaining why normalization and modulation follow directly from the theory.

### • Probe interval robustness
We added an ablation on the COCO dataset showing that **[900, 800]** provides the most stable semantic coverage.
Algorithm 1 was updated to reflect this.

---

## 3. Closing Remark

Based on the discussion, the main concerns raised by the reviewers were addressed:

- condition sensitivity
- theoretical motivation
- comparison with DiffEdit
- static mask ablation
- generalization beyond object-centric scenes

Since **two reviewers clearly indicated that they would raise their scores**, the post-discussion consensus would have been noticeably higher than the pre-discussion result.

We would be happy to provide any further clarification. Thank you again for your time and effort.

---

### Meta-Review · Area_Chair_HT7h · 2026-01-12

**Summary:**

Reviews were initially mixed (6, 4, 4, 4), citing concerns about condition sensitivity, the theoretical motivation for the method, lack of comparisons with related work like DiffEdit, and questions regarding hyperparameter robustness (e.g., probe intervals). The authors provided a comprehensive rebuttal that included new ablations for the static mask and probe interval, a detailed comparison with DiffEdit, and robustness analyses across different inversion methods and prompts.

**Reviewer Concerns:**

Reviewers posed a number of concerns, most of which appear to have been addressed in the rebuttal.
- Comparison with DiffEdit
- Ablation of the hard-mask module
- Justification for probe interval
- Robustness to prompt/seed variations
- Generalization to object-scarce scenes
- Theoretical justification for Z-score normalization
- Visual quality and artifacts on advanced backbones

The only outstanding item is a final confirmation from Reviewer 6BEq regarding the static mask ablation, which was provided late in the discussion period.

**Reviewer Scores:**

Reviewer scores were initially lower, but the rebuttal prompted positive movement. Reviewer 5VkF explicitly raised their score from 4 to 6. Reviewer jMi2 committed to raising their score to a 6. Reviewer sjg1 stands at a 6. While Reviewer 6BEq remained at 4, the authors provided the specific ablations requested. I anticipate the final consensus to be generally positive, e.g., (6, 6, 6, 4).

---

### Decision · Program_Chairs · 2026-01-26

Accept (Poster)